



# Optical Detection of Magnetic Resonance

Dieter Suter

Experimental Physics III, TU Dortmund University, 44227 Dortmund, Germany

**Correspondence:** Dieter Suter (Dieter.Suter@tu-dortmund.de)

**Abstract.** The combination of magnetic resonance with laser spectroscopy provides some interesting options for increasing the sensitivity and information content of magnetic resonance. This review covers the basic physics behind the relevant processes, such as angular momentum conservation during absorption and emission. This can be used to enhance the polarization of the spin system by orders of magnitude compared to thermal polarisation as well as for detection with sensitivities down to the

level of individual spins. These fundamental principles have been used in many different fields. This review summarises some of the examples in different physical system, including atomic and molecular systems, dielectric solids composed of rare earth and transition metal ions and semiconductors. [1]

## 1 Introduction and Overview

### 1.1 Sensitivity of Magnetic Resonance

Magnetic resonance spectroscopy basically measures the interaction of electronic or nuclear angular momenta with each other and with external magnetic fields (Abragam, 1961). While the fundamental processes that occur during magnetic resonance have been understood for at least seventy years, the field has continued to expand in many directions, mostly due to the ever increasing possibilities to use spins as probes of their environment. Today, the biggest remaining weakness of the technique is its relatively low sensitivity, compared, for example, to optical experiments. In the area where magnetic resonance spectroscopy

has become most popular, that of nuclear magnetic resonance (NMR) of liquids, the minimum number of spins that can be detected is of the order of $10^{17}$. In contrast to this, the ultimate sensitivity limit, i.e. spectroscopy of individual particles has been demonstrated for optical systems four decades ago (Neuhauser et al., 1980).

Several issues contribute to this low sensitivity. The main reason is that the interaction energies are relatively small ($< 10^{-22}$ J), so that the corresponding frequencies are in the radio-frequency (RF) or microwave (MW) regime ($< 10^{11}$ Hz). The small

interaction energy results, e.g., in small thermal population differences between the energy levels participating in a particular transition, which are determined by the Boltzmann factor $e^{-h\nu/k_B T}$. For optical transitions, where $\nu$ is of the order $10^{18}$ Hz, the population ratio is thus close to unity, while for RF transitions ($\nu < 10^9$ Hz), they are of the order of $10^{-4}$. Similarly, the energy of an RF photon is, under typical conditions, well below the thermal noise level. This makes it very difficult to detect a single RF photon, in contrast to optical photons, which can be detected with efficiencies close to unity.

---

[1]This review was originally written in response to an invitation of "Progress in NMR Spectroscopy" but re-directed to "Magnetic Resonance" to be accessible to a wide audience. This paper has been reviewed by peers in accordance with the policy of "Magnetic Resonance".





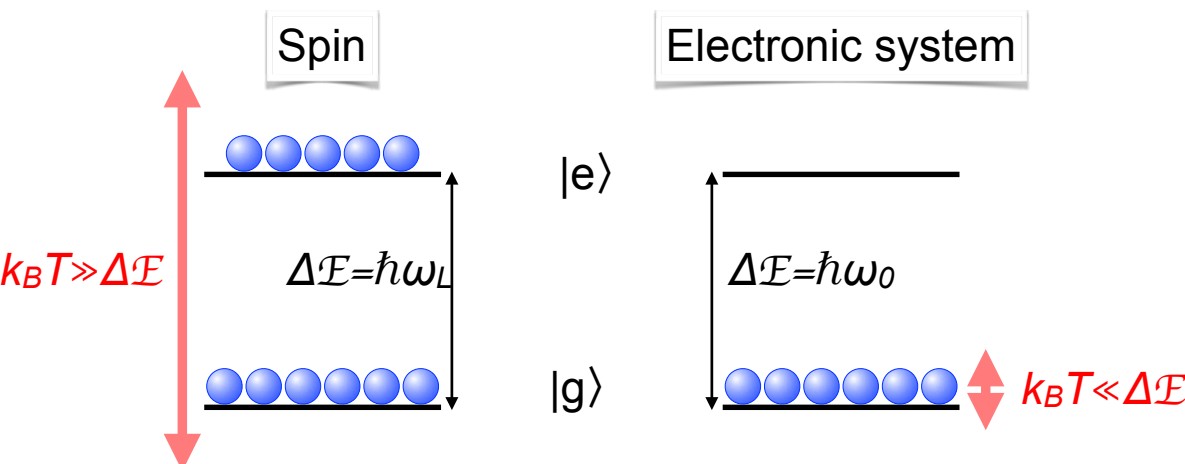

**Figure 1.** Comparison of spin- and optical transitions with the corresponding populations at room temperature.

## 1.2 Optics and lasers

While magnetic resonance is concerned with transitions between states that differ in terms of their spin quantum numbers, optical transitions occur between states that differ in terms of their electronic confiuguration. The energy difference between these states is typically of the order of electron Volts (eV), which corresponds to transition frequencies of $\approx 4 \cdot 10^{14}$ Hz. Since this energy difference is large compared to the thermal energy of the system, only the lowest states are populated in thermal equilibrium. Figure 1 visualises some of these differences.

Early experiments on optical excitation and detection of magnetic resonance used conventional light sources such as discharge lamps. These light sources had a very limited intensity, which implied that the systems with which they interacted, were only weakly perturbed. A significant effect, e.g. in terms of establishing a significant population difference between the spin states, could only be achieved if the relaxation processes could be kept to a minimum. In those early experiments, light was used mainly in order to polarize the spin system and to observe the precessing magnetization, while RF irradiation was used to change the dynamics of the spin system. Nevertheless, it was realized early (Cohen-Tannoudji, 1962; Cohen-Tannoudji and Dupont-Roc, 1972) that optical radiation cannot only polarize the spin system, but also leads to shifts and broadening of the magnetic resonance transitions. With the introduction of the laser, the available light intensity and the coherence properties of the radiation field changed in such a way that many experiments that had not been feasible before have become routine (Demtröder, 1991; Shen, 1984; Balian et al., 1977). One important example is the generation of ultrashort laser pulses, which provide high intensity as well as high time resolution.

The advent of the laser also led to the revisions on the theoretical side. In particular, the high spectral purity and large intensity of the laser light result in a nonlinear response of the system to the optical field and to additional phenomena, such as selective excitation. In many cases, the optical coherences have to be taken into account, and the dynamics must be formulated in terms of the density operator (Decomps et al., 1976). Other effects, which were discovered with discharge lamps, but were




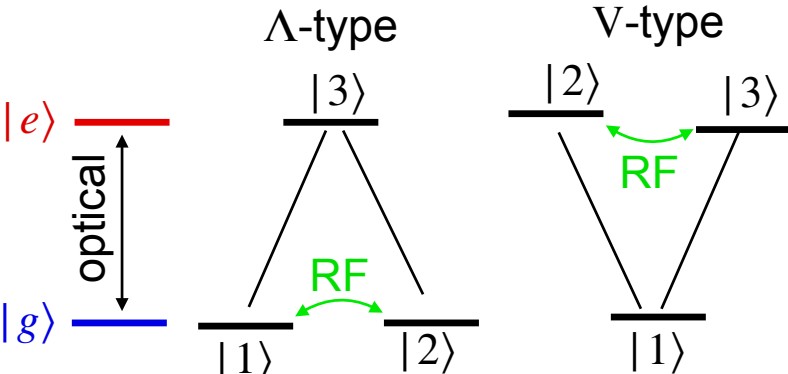

**Figure 2.** The most important 3-level systems that allow basic optical excitation and detection of magnetic resonance. The arrows indicate the transitions coupling to optical and RF photons.

too small to be of practical significance, were increased by many orders of magnitude. For example, the light-shift effect, an apparent shift of energy levels due to optical irradiation of an adjacent transition have the same effect on the spin dynamics as magnetic fields. By selectively irradiating certain optical transitions, these virtual magnetic fields can be used as an additional degree of freedom for the modification of spin dynamics. It is therefore possible to perform many experiments by purely

optical methods; the usage of the optical radiation field is then threefold: it polarizes the spin system by transferring angular momentum from the photons to the spin system, it modifies the dynamics of the system via an effective Hamiltonian, and it is used to detect the resulting time-dependent magnetization,

### 1.3   Coupled optical and magnetic resonance transitions

While the systems under study can have very different energy level schemes, the basics of the techniques can often be explained
in terms of a simple three- level scheme (Fig. 2). The transition of interest is between the two spin-substates of a given electronic state, i.e. between the two ground state in the case of the $\Lambda$-type system and between the two excited states in the $V$-type system (right-hand part of Fig. 2). In many actual cases, both types of transitions occur in the same system, so that resonances in both the ground and excited states can be excited.

### 1.4   Optical pumping and optical detection

In cases where the sensitivity provided by classic magnetic resonance is not sufficient, it is often possible to increase the population difference between the different magnetic sublevels by optical pumping (see, e.g., (Balling, 1975; Bernheim, 1965)). Like the Population difference between ground and electronically excited states, the population difference between levels differing only in their spin state can then reach values near unity.

    Conversely, the population difference and coherence between the magnetic substates can change the optical properties of the
system; it is therefore possible to detect the magnetization optically, with a sensitivity much greater than if the radio frequency





photons are detected (Kastler, 1967; Bitter, 1949). In simple cases, this gain in sensitivity can be understood as an amplification of the radiation by transferring the angular momenta from the internal degrees of freedom of the system to photons with optical energies instead of RF energies. In classical terms, this transfer of angular momentum basically leads to a (circularly) polarized radiation field.

## 1.5 Motivation

Optical techniques have turned out to be useful in many different areas of magnetic resonance. As pointed out above, the main motivation is often the gain in sensitivity, which also has allowed us to reach the single-spin limit (Wrachtrup and Finkler, 2016; Aslam et al., 2017). Apart from the gain in sensitivity, the use of optical radiation also provides the option to perform magnetic resonance spectroscopy of electronically excited states. Since these states are not populated in thermal equilibrium, the targeted systems must be brought into the excited state before magnetic resonance can be performed. If the excitation can be achieved with light, it is often advantageous to use selective excitation of the magnetic substates to obtain a spin-polarized system. This is also necessary since the excited state population that can be achieved may be substantially smaller than in the ground state so that sensitivity again becomes an important issue. The fluorescence emitted by these systems, is often polarized and can be used directly to measure the excited state magnetization.

A third reason to combine magnetic resonance with optical techniques is that the information content of double-resonance experiments is often higher than the information that can be obtained with the individual techniques. This includes, e.g, the identification of fluorescent centers through their magnetic resonance spectra, selective excitation and detection of magnetic resonance near a surface (Grafström and Suter, 1995; Grafström et al., 1996; Grafström and Suter, 1996a, b) or in selected semiconductor quantum wells (Eickhoff et al., 2003). In some cases, the double resonance experiment allows one to break symmetries inherent in magnetic resonance. Breaking them in a controlled way can allow one to differentiate between positive and negative signs in some coupling constants (Nilsson et al., 2004) or to obtain orientational information from an isotropic medium such as a frozen solution (Börger et al., 2001).

## 2 Physical Background

Combining optical methods with magnetic resonance is possible if a system has electronic as well as spin degrees of freedom. Optical fields generally interact with transitions between different electronic states through the electric dipole interaction, while magnetic resonance drives transitions between states that differ with respect to their spin (or, more generally, angular momentum) degrees of freedom. The two subsystems are often coupled, since the optical transitions connect states that differ not only with respect to their electronic configuration, but also in terms of their angular momentum.

### 2.1 Angular momentum and selection rules

Spins are an important form of angular momentum and magnetic resonance is the main approach to excite and detect transitions between states that differ only in angular momentum. In the case of nuclear magnetic resonance, the spin angular momentum



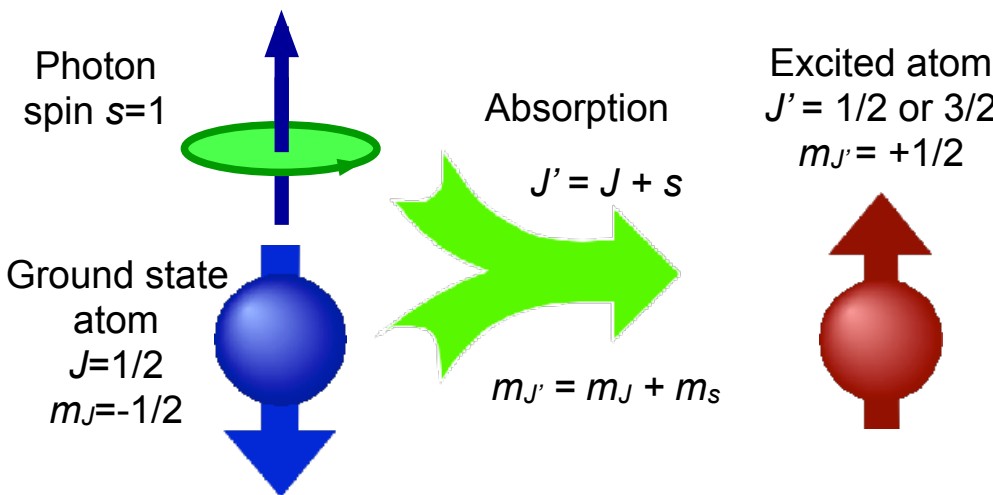

**Figure 3.** A simple absorption process illustrating angular momentum conservation in atomic physics.

of atomic nuclei is the object under investigation, while in the case of EPR, the spin of the electron can be mixed with orbital angular momentum of the electron. Radiation also carries angular momentum. While it is possible to determine the angular momentum of an electromagnetic wave in terms of Maxwell's equations, it becomes much more relevant in quantum mechanics, where each photon carries a spin angular momentum of $\hbar$. In vacuum, it is sufficient to consider two of the three angular momentum states, e.g., the cases where the spin is oriented parallel to the direction of propagation, and opposite to it. These states correspond to circularly polarized light.

If the environment of the object being studied is isotropic, such as in a free atom, angular momentum is a conserved quantity. Accordingly, a change of the spin state must be compensated by a change of some other form of angular momentum. This is important, e.g., during absorption or emission of photons: The angular momentum of the photon that is created or destroyed must be compensated by a corresponding change of the angular momentum state of the system that absorbs (or emits) the photon.

Figure 3 shows this exchange of angular momentum between matter and radiation field for a simple case taken from atomic physics. In this example, the electronic ground state as well as the electronically excited states have an electronic angular momentum of $J = \hbar/2$. This corresponds, e.g., to the $D_1$ line of alkali atoms, where the ground state angular momentum is given by the spin $S = \hbar/2$ of the electron, while its orbital angular momentum vanishes. The first excited state has an orbital angular momentum of $L = \hbar$ and is split into the $J = L \pm S$ states $^2P_{3/2}$ and $^2P_{1/2}$, which are connected to the ground state by the $D_2$ and $D_1$ lines. If the atom absorbs a photon, it must change its internal state such that the angular momentum $\boldsymbol{J}'$ of the final state of the atom is equal to the sum of the initial angular momentum $\boldsymbol{J}$ of the atom plus the spin angular momentum $\boldsymbol{s}$ of the photon, $\boldsymbol{J}' = \boldsymbol{J} + \boldsymbol{s}$.



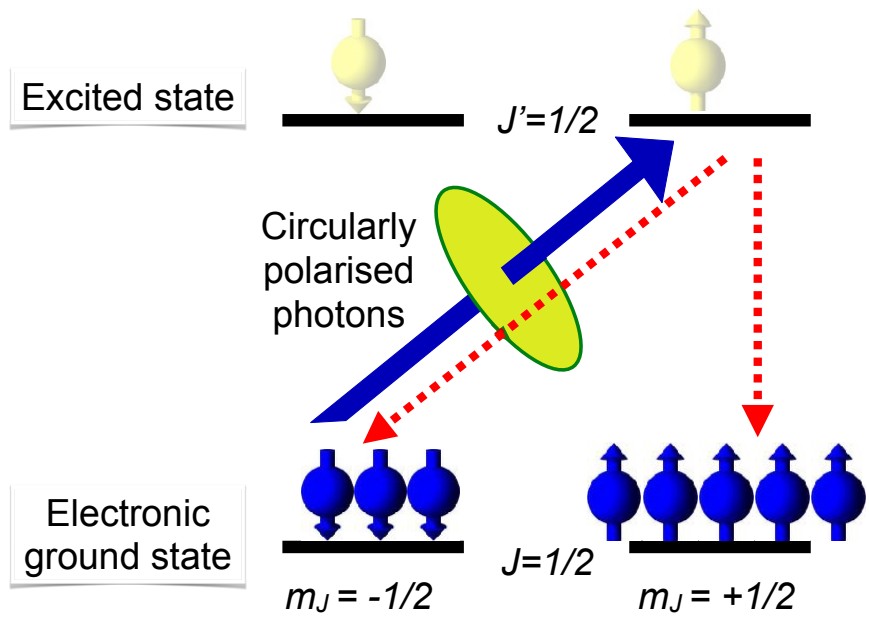

**Figure 4.** Basic principle of optical pumping. The solid arrow indicates the only transition that couples to a circularly polarised laser field in this level system. The dashed arrows indicate transitions for spontaneous emission.

## 2.2 Optical Pumping

Exciting spins in magnetic resonance requires that the states do not all have the same occupation probability. In conventional magnetic resonance experiments, thermal contact of the spins with the lattice establishes the polarization. This process is relatively slow, especially at low temperatures where relaxation times can be many hours. In addition, the polarization is limited by the Boltzmann factor, which is typically less than $10^{-4}$. Much higher polarizations can be achieved by transferring population differences from photons, which can easily be prepared in pure states, i.e. with a polarization of 100 %. This transfer process is known as optical pumping(Kastler, 1967).

Figure 4 illustrates the basic process of optical pumping for a simple 4-level system. In this example, the electronic ground state as well as the electronically excited state have total angular momentum $J = 1/2$. If the laser field is circularly polarised, it contains only photons whose angular momentum $J_p = 1$ (in units of $\hbar$) is oriented parallel to the direction of propagation. The system that absorbs the photon has to accommodate its energy as well as its angular momentum. The energy is absorbed by changing from the electronic ground state to the electronically excited state, while the angular momentum is absorbed by changing from the $m_J = -1/2$ to the $+1/2$ state.

Since the excited state is not stable, the system falls back into nthe ground state. During this process, the absorbed photon energy is dissipated either as a photon (radiative relaxation) or by transfer to other degrees of freedom (radiation-less). In both cases, the system can end up in either of the two ground states. Since the $|g, \uparrow\rangle$ state does not couple to the laser field, the population of this state grows continuously, if the process is repeated, and the system can be pumped into this state.





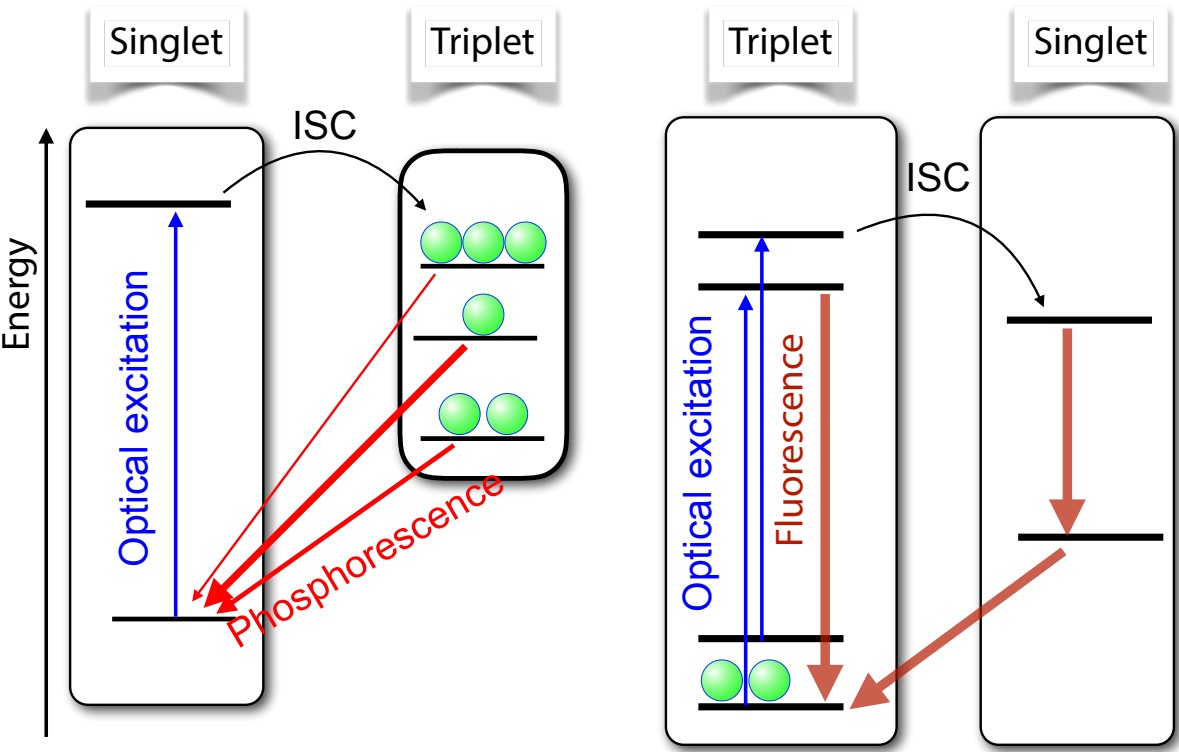

**Figure 5.** Basic principle of spin-selective intersystem crossing (ISC) for the case of a singlet ground state (left) and a triplet ground state (right). In both cases, the optical irradiation results in unequal populations of the triplet states.

While this simplified level scheme is quite useful for understanding the basic processes occuring during optical pumping, it is important to consider real systems for any quantitative analysis. In particular, the presence of a nuclear spin has very significiant effects, aspart from the existence of hyperfine splitting. As an example, the optical pumping process in the true atomic ground state is nonexponential and slower by at least an order of magnitude, compared to a hypothetical atom with a vanishing nuclear spin (Suter, 1992).

### 2.3 Other Mechanisms for Optical Polarisation

The basic mechanism for optical pumping described above is easily understood in terms of angular momentum conservation. In solid-state systems, however, space is not isotropic and angular momentum is in general not a preserved quantity. In such systems, it is sometimes possible to obtain polarisation of electronic as well as nuclear spins via different optical pumping processes.

An important example is the case of spin-selective intersystem-crossing (ISC), where electronic singlet states convert to triplet states and vice versa. Figure 5 shows the basic principle for the two most important cases. On the left-hand side, the ground state is a singlet state. A laser photon brings it to an excited singlet state, which has a finite probability to make a





radiationless transition to a triplet state. The interactions between the unpaired electron spins in the triplet state and their spin-orbit interaction give rise to the zero-field splitting which completely or partially lifts the degeneracy of the spin states. The ISC transition rates from the singlet to the different triplet states depend on the spin-orbit coupling and can therefore differ significantly, resulting in different populations of the triplet states. Similarly, the triplet state lifetimes are in general different,

as indicated by the different thickness of the arrows that connect the triplet states to the singlet ground state. These different lifetimes again lead to unequal populations of the three triplet states.

If a resonant microwave field drives one of the transitions between the triplet states, this affects the populations and therefore also the photon emission rates. Accordingly, this system is quite well suited for measuring the energy differences between excited triplet levels via optical detection. In addition, the polarisation of the electron spin can be transferred to coupled nuclear

spins. The resulting nuclear spin polarisation survives the transition into the singlet ground state, where it can accumulate over multiple absorption - emission cycles.

The right-hand part of Figure 5 shows a different type of system, where the ground state is a triplet state. Absorption of laser photons and direct emission of photons in zero magnetic field is in general spin-preserving. However, the different spin states can have vastly different probabilities of undergoing ISC to the singlet state. In the important case of the nitrogen-vacancy

(NV) center in diamond, this allows one to pump most of the electron spin population into the $m_S = 0$ state of the electronic ground state (Doherty et al., 2013; Suter and Jelezko, 2017).

Examples where this mechanism generates high spin polarisation include organic molecules like pentacene (Kothe et al., 2010; Iinuma et al., 2000) or quinoxaline(von Borczyskowski and Boroske, 1978), oxygen-vacancy complexes in silicon (Itahashi et al., 2013) and the NV-center in diamond (Doherty et al., 2013; Suter and Jelezko, 2017; Ajoy et al., 2018a, b; Zangara

et al., 2019; Chakraborty et al., 2017). Similar processes are responsible for the polarisation of spins in quartet states that undergo ISC to doublet states, such as some defects in SiC (Baranov et al., 2011).

## 2.4 Optical Detection

Classical magnetic resonance relies mostly on the detection of time-dependent magnetisation by coupling an oscillating component of the associated magnetic flux to an external antenna like a coil or a microwave cavity. In most cases, this coupling can

be well described by Faraday's law of induction or, equivalently, Maxwell's third law. The optical detection techniques that are discussed in this section, however, do not depend on magnetic flux. Instead, the angular momentum (mostly spin) couples directly or indirectly to the angular momentum of some optical pohotons or to other degrees of freedom of the optical field. The possibility of using optical properties for detecting magnetic resonance was first suggested by Bitter (Bitter, 1949).

In the following, we start with a relatively general discussion, where the spins can be either electronic or nuclear spins, and

they may be located in solid or gaseous samples (liquids are less suitable for this type of experiments). We will therefore refer to them with the general term 'particles', which will stand for all types of spin-carrying centers under study.





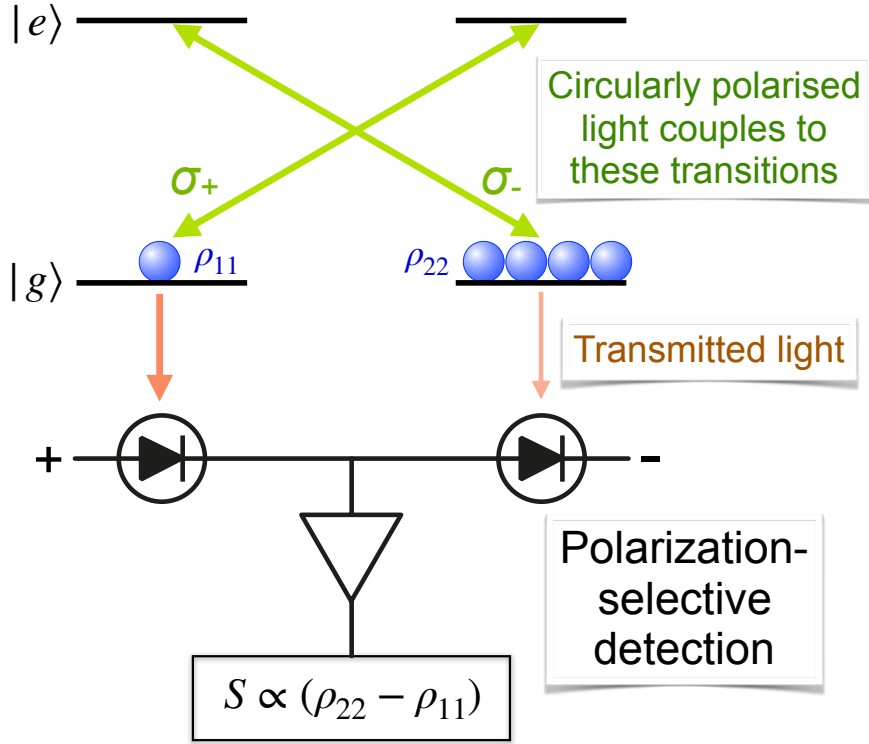

**Figure 6.** Basic principle of optically detecting magnetic resonance in transmission by differential absorption.

### 2.4.1 Absorption / Transmission

Figure 6 illustrates a simple mechanism for optical detection: Light with a given circular polarization interacts only with one of the transitions between the ground state $|g\rangle$ and the excited state $|e\rangle$. We assume for simplicity that all particles are in the electronic ground state, but the two spin states have different populations. Since the absorption of the medium is directly proportional to the number of atoms that interact with the light, a system with spin polarisation is circularly birefringent and dichroic, i.e. the refractive index and absorption coefficients for the two opposite circular polarisations are different. Comparison of the absorption or dispersion of the medium for the two opposite circular polarizations yields a signal that is proportional to the population difference and thus to the spin polarisation along the direction of propagation. This type of measurement is often used, e.g., in different forms of experiments in ultra-low magnetic field. Some examples are discussed in section 3.1.

In the ideal situation, where a particle in a spin state $|\uparrow\rangle$ ($|\downarrow\rangle$) absorbs only left (right) circularly polarised light, a beam of light that has initially the same number $n_{p\pm}(0)$ of photons for both circular polarizations, undergoes differential absorption. Behind the sample of length $\ell$, the numbers are

$$n_{p\pm}(\ell) = n_{p\pm}(0)e^{-\ell\alpha_0 N_0 p_\pm},$$





where $\alpha_0$ is the absorption coefficient per particle, $N_0$ the total particle density, and $p_\pm$ is the fraction of spins in the states $\uparrow/\downarrow$. If the exponent is small, we can use a linear expansion

$$n_{p\pm}(\ell) = n_{p\pm}(0)(1 - \ell\alpha_0 N_0 p_\pm).$$

Difference detection, as shown in figure 6, for initially equal photon numbers, $n_{p+}(0) = n_{p-}(0) = n_0$, yields

$$\Delta n(\ell) = -2n_0\ell\alpha_0 N_0(p_+ - p_-) = -2n_0\ell\alpha_0 N_0 \Delta p, \tag{1}$$

where we have used the fractional spin polarisation $\Delta p = p_+ - p_-$. Since difference detection is free of background, this signal is not significantly perturbed by classical noise of the laser. It is, however, affected by shot noise, which was not considered in this classical analysis. According to eq. (1), a high sensitivity (i.e. small $N_0$ and $\Delta p$) can be achieved by using a large $n_0$ (i.e. high laser intensity), a long path length $\ell$ and a large absorption coefficient $\alpha_0$. These goals tend to be incompatible, however. As an example, large absorption coefficients and long path lengths lead to an inhomogeneous system and violates the assumption of linearity made in this derivation, while the combination of high laser intensity and large absorption coefficient lead to unwanted perturbations of the system. In cases where these issues become important, it is possible to modify the basic scheme discussed above, e.g. by using dispersive instead of absorptive detection (Suter et al., 1991). In this case, the complex index of refraction (i.e. absorption as well as dispersion)(Rosatzin et al., 1990) depend linearly on the spin polarisation of the ground state.

### 2.4.2 Spontaneous Emission

When electronically excited states are populated during an experiment, their return to the ground state may be accompanied by the emission of a photon that carries information about the state that was populated. While it is much harder to detect spontaneously scattered photons, since they are emitted over a large solid angle, they provide a significantly higher information content than the transmitted laser photons: they are all emitted (if correctly filtered) by the system under study.

Whether these photons are actually useful depends on the system. Figure 7 shows a simple but important case: If the environment of the emitter has sufficiently high symmetry, such as in the case of free atoms, angular momentum conservation requires that the angular momentum of the photon is equal to the difference between the angular momenta of the two atomic states,

$$s = J_e - J_g.$$

Here, $s$ is the photon angular momentum, while $J_{e,g}$ are the angular momenta of the electronically excited and the electronic ground state of the atom.

In systems with lower symmetry, the angular momentum may not be a conserved quantity, and the polarisation of the photons may not depend on the spin of the participating states. Even in those cases, however, it may be possible to infer the angular momentum state of the quantum system from some properties of the measured fluorescence. A good example is the NV-system in diamond, which will be discussed in section 4.3. Here, the number of scattered photons is a good indicator of the angular




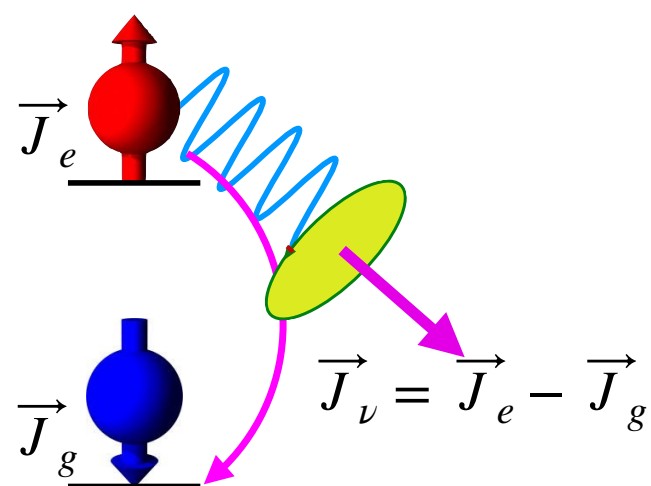

**Figure 7.** Angular momentum conservation during photon emission.

momentum state: If the system is initially in the $m_S = 0$ state, the photoluminescence rate is typically 20% higher than for the $m_S = \pm 1$ states (Doherty et al., 2013; Suter and Jelezko, 2017).

Changes in the rate of spontaneous emission can not only be induced by driving spin transitions with MW or RF fields, but also by tuning the energy levels with a static magnetic field. As an example, a magnetic field can tune the energy of long-lived states (e.g. due to a spin-forbidden transition to the ground state) to match the energy of a state with a short radiative lifetime. As a result, even small symmetry-breaking terms mix the two (near-)degenerate states, resulting in significant increase of the photoemission rate and/or the polarisation of the PL. Since the coupling terms mix the two levels, the degeneracy is avoided and the system goes through a level anticrossing (LAC). Measuring these resonances (see, e.g., (Baranov and Romanov, 2001))

corresponds to a magnetic resonance experiment without an alternating (ac) field.

### 2.4.3   Coherent Raman Scattering

Scattering of light by spin systems cannot only occur spontaneously, but also in a coherent manner. A good example is coherent Raman scattering, where a coherent electromagnetic field (typically a laser field) interacts with the system under study to generate a second field, whose frequency and possibly momentum and polarisation are different from the incident field. While

the cross sections of such processes may be relatively small, the scattered field depends strongly on the system properties and therefore carries a significant amount of information about the system.

    Figure 8 illustrates the process for a basic 3-level system, where a superposition of the two ground state levels $|1\rangle$ and $|2\rangle$ corresponds to a precessing magnetization, which can, in principle, be detected via the voltage induced in an RF coil. In the case of detection through a laser field, the incident laser is (near-) resonant with transition $|1\rangle \leftrightarrow |3\rangle$. It therefore generates a



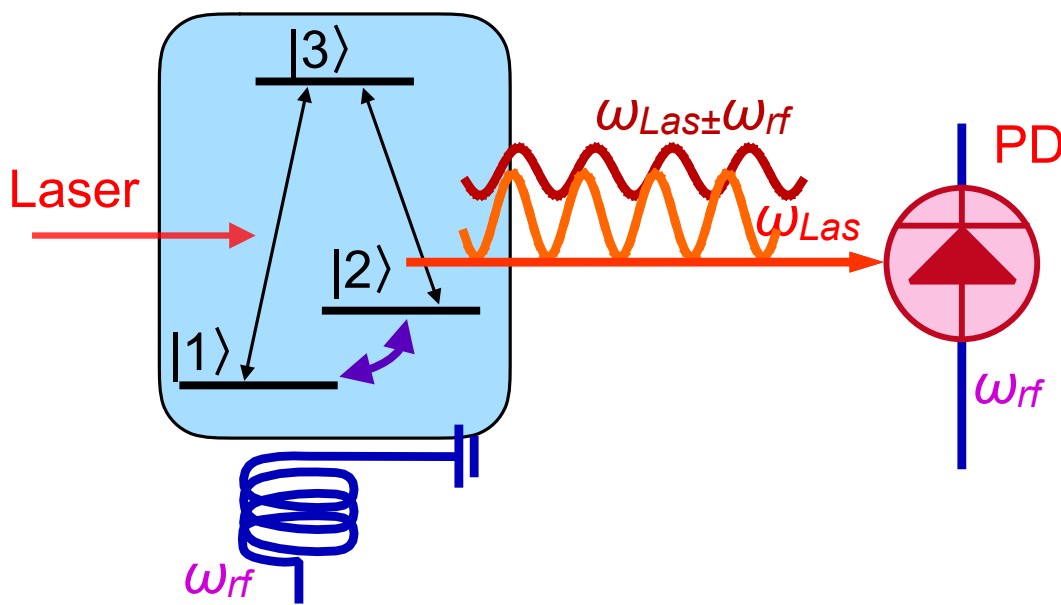

**Figure 8.** Basic principle of coherent Raman scattering.

coherent superposition of these two levels

$$|1\rangle \xrightarrow{\text{Laser}} \frac{1}{\sqrt{2}}(|1\rangle + |3\rangle),$$

which will then evolve as

$$\frac{1}{\sqrt{2}}(|1\rangle + |3\rangle e^{-i\omega_{13}t}),$$

where $\omega_{13}$ is the energy difference between the states $|3\rangle$ and $|1\rangle$, divided by $\hbar$. This coherent superposition state corresponds to an electric dipole oscillating at the laser frequency $\omega_L$. Due to the coherence between levels $|1\rangle$ and $|2\rangle$, however, it simultaneously generates coherence in the transition $|2\rangle \leftrightarrow |3\rangle$,

$$\frac{1}{\sqrt{2}}(|1\rangle + |2\rangle) \xrightarrow{\text{Laser}} \frac{1}{2}|1\rangle + \frac{1}{\sqrt{2}}|2\rangle + \frac{1}{2}|3\rangle,$$

which represents a coherent superposition of all 3 states and therefore contains coherence in all 3 transitions. It evolves as

$$\frac{1}{2}|1\rangle + \frac{1}{\sqrt{2}}|2\rangle e^{-i\omega_{12}t} + \frac{1}{2}|3\rangle e^{-i\omega_{13}t}.$$

The coherence in the three transitions therefore oscillate with frequencies $\omega_{12}$, $\omega_{13}$, and $\omega_{23} = \omega_{13} - \omega_{12}$. While $\omega_{12}$ is in the RF or MW range, the frequencies $\omega_{13}$ and $\omega_{23}$ are optical frequencies. Accordingly, the system emits two optical fields with these frequencies, with optical polarizations that are determined by the dipole moments of the two transitions.

While this process shares properties of optical and magnetic resonance, it also shows some unique features. As an example, in a CW experiment, where the transitions $|1\rangle \leftrightarrow |2\rangle$ and $|1\rangle \leftrightarrow |3\rangle$ are irradiated continuously, the amplitude

$$s_{23} \propto \mu_{ij}\mu_{jk}\mu_{ki}$$





of the emitted field in the transition $|2\rangle \leftrightarrow |3\rangle$ is given by the product of all three transition amplitudes (Wong et al., 1983; Neuhaus et al., 1998). One consequence of this tri-linearity is that in many occasions, there is interference between different signal contributions, which can lead to cancellations (Kintzer et al., 1985; Mitsunaga et al., 1984, 1985). In particular, many systems show a symmetry between the Stokes scattering pathway, where the frequency of the scattered field is equal to the difference $\omega_L - \omega_{RF}$ between the frequency $\omega_L$ of the laser field and the RF frequency $\omega_{RF}$ and the anti-Stokes field, whose frequency is the sum $\omega_L + \omega_{RF}$. If these signals have opposite amplitudes, they cancel and the expected resonance line appears to be missing. This can be avoided by suppressing the symmetry, e.g. by adding a pump laser beam that reduces the population difference across one of the 2 transitions or increases that of the second transition.

### 2.5 Sensitivity limits

As discussed above, optical methods can increase the population difference of spin systems by many orders of magnitude and they increase the detection sensitivity compared to inductive detection. This is not only due to the higher signal energy of optical photons, but also to the virtual absence of thermal noise at optical frequencies. A third reason for the increased sensitivity is that laser irradiation can polarize the spins much faster: depending primarily on the laser intensity, complete polarization of the spin system may require less than 1 $\mu$s (Suter and Jelezko, 2017). Since optical detection directly measures the magnetization, in contrast to pick-up coils that measure its time derivative, the detection sensitivity is independent of the resonance frequency. It is therefore possible to perform experiments at low or vanishing fields with the same detection efficiency as at high fields.

#### 2.5.1 Transmission

In a transmission experiment described in section 2.4.1, the relevant noise has mostly three contributions: (i) laser intensity fluctuations, (ii) shot noise of the laser beam and (iii) thermal noise of the detection. The first type of noise can be reduced significantly e.g. by the balanced detection scheme shown in Fig. 6. The thermal noise of the detector can be reduced by technical measures, such as cooling. However, it is not always possible to reduce this contribution to insignificant levels. The shot noise of the laser, finally, is not elminated by the balanced detection scheme, since it is anti-correlated between the two channels. Keeping this contribution low, relative to the signal, requires one to use high laser power, which is naturally limited by the properties of the sample: not only overheating can be a problem, but also the dynamics of the system that one tries to detect may be modified at high laser intensity.

#### 2.5.2 Fluorescence

The sensitivity of fluorescence detection varies significantly between systems. Ideally, it would be possible to determine the spin state of a particle by measuring the polarisation of a single fluoresence photon. Such a measurement could be repeated immediately after detecting the photon, which allows a readout with very high certainty in a time below one microsecond. In real systems, however, the actual readout time is significantly longer. In the case of a diamond NV-center, e.g., the rate at which photons are detected is typically of the order of $10^5 \text{s}^{-1}$. Furthermore, the information is carried here only by the rate at which




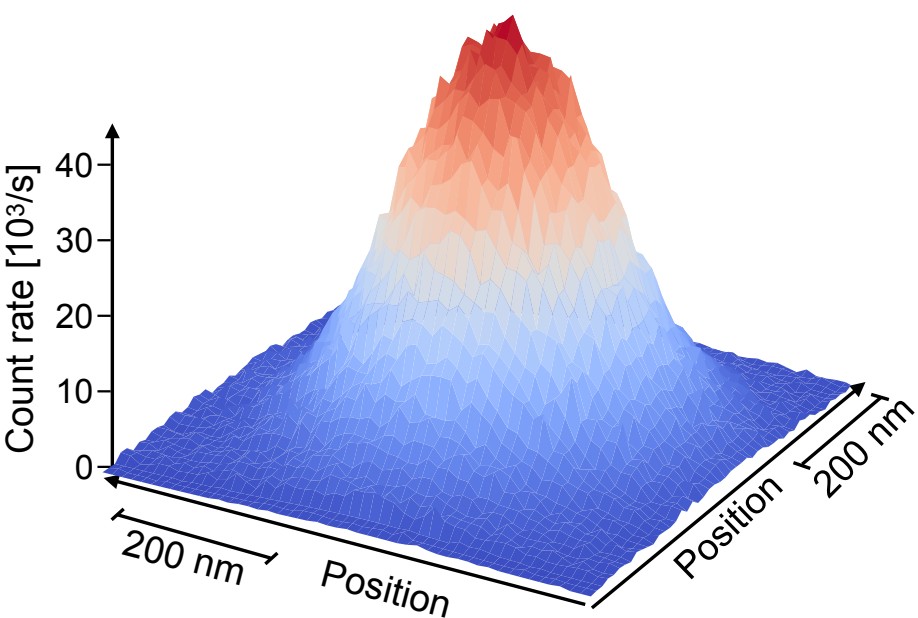

**Figure 9.** Photon count rate from a single NV center in diamond measured as a function of position.

photons are emitted (not by the polarization) and this rate differs between the spin states by some 20% (Doherty et al., 2013; Suter and Jelezko, 2017). Accordingly, it typically takes up to 1 ms to determine the state of a single spin with sufficiently high certainty.

### 2.5.3 Single spin detection

Detecting the signal of single spins (Wrachtrup and Finkler, 2016) mostly requires the suppression of unwanted signal contributions from other sources, such as scattered laser light or fluorescence from other sources. Consider, e.g. the common situation of a NV center in diamond, where a 0.1 mW laser beam is used for excitation, which corresponds to some $2.7 \cdot 10^{15}$ photons per second. From a single center, we typically obtain some $10^5$ fluorescence photons per second. Accordingly, the fraction of laser photons that can be allowed to reach the detector must be less than $\approx 10^{-12}$ to avoid significant degrading of the signal. This is typically achieved by a combination of spectral filtering, where the higher-energy laser photons are reflected or absorbed, while the lower-energy fluorescence photons are transmitted, and spatial filtering, e.g. by the pinhole of a confocal microscope.

Fig. 9 shows the photon count rate from a single NV center measured as a function of position. The width of the peak (HWHH), which is determined by the resolution of the instrument, is 113 nm. This diffraction-limited focus is still large compared to the size of the center ($\approx 0.1 \, \text{nm}$), but it suppresses background signals from the rest of the sample and allows therefore to detect a single center by accumulating photons for $\approx 1 \, \text{ms}$. Other examples of experiments with single spins are discussed in the subsequent sections.





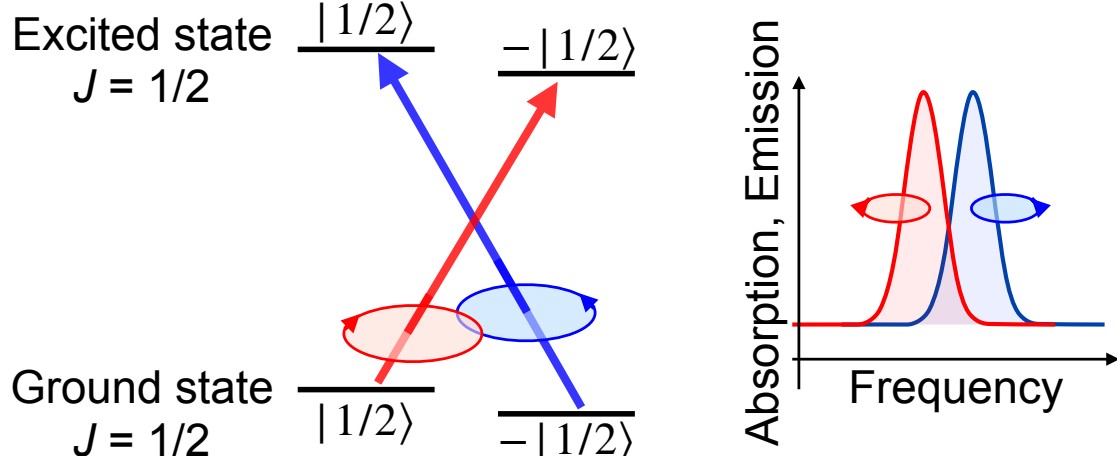

**Figure 10.** Energy level system and resonance lines of a Zeeman doublet.

## 3 Atomic and molecular systems

Atomic vapors are very useful systems for studying the basics of optically detected magnetic resonance. Very often it is possible
to neglect spatial degrees of freedom and interactions between atoms. In these cases, the Hilbert space of the system can be
limited to the angular momentum degrees of freedom plus a small number of electronic states that interact with the laser beam.
We therefore start the discussion with these systems and move to more complicated systems with higher practical relevance in
the following chapters.

### 3.1 Detection of electron spin

Perhaps the most direct possibility for detecting magnetic resonance optically consists in using a high-resolution optical spectrometer and monitor the intensity of two resonance lines from a Zeeman doublet.

Figure 10 shows the basic principle: The system consists of an electronic ground state and an electronically excited state,
both of which can have two possible angular momentum states. As discussed in section 2, optical transitions can only occur
between pairs of states that allow conservation of angular momentum; In the example shown here, circularly polarised light
interacts either with the blue or the red transition. A magnetic field shifts the ground- as well as the electronically excited states
in different directions and therefore lifts the degeneracy of the resonance lines. In the example shown in the figure, the red transition is shifted to lower frequencies while the blue transition shifts to higher frequencies. If a magnetic resonance experiment
changes the relative occupation numbers of these angular momentum states, it affects the amplitudes of the resonance lines in
different ways. A resonant excitation of the spin transitions can therefore be monitored through the change in the amplitudes
of the resonance lines or, if they are not completely resolved, through an asymmetry in the spectrum or a shift of the mean
frequency (Geschwind et al., 1965).





In many systems, even the natural linewidth of the relevant transitions is large compared to the difference in their resonance frequencies, so the overlap is too large to allow for their separation in a spectrometer. Nevertheless, it is possible to monitor differences in the populations of the spin substates through their effect on absorption, dispersion and emission of optical transitions between these states. As discussed in section 2.4.1, the polarisation of light transmitted through a resonant medium depends on the spin polarisation in the medium. It can thus be used to monitor the spin polarisation.

Atomic vapors, particularly of alkali metals, are the simplest systems for discussing and testing these effects. It is therefore not surprising that they were also the first systems where the effect was studied (Kastler, 1967; Lange and Mlynek, 1978; Suter and Mlynek, 1991). In this case, the complex index of refraction (i.e. absorption as well as dispersion)(Rosatzin et al., 1990) depend linearly on the spin polarisation of the ground state. It is therefore possible to measure a component of the spin polarisation by transmitting a laser beam in that direction through the sample and detecting either the circular dichroism or the Faraday rotation of the light - typically with differential detection schemes like those discussed in section 2.4.1.

Such experiments can not only probe the bulk of an atomic vapor, they can also be used to selectively study interfaces, e.g. by reflecting a laser beam from an interface between glass and vapor (Suter et al., 1991; Grafström and Suter, 1995) or from the surface of a crystal (Lukac and Hahn, 1988), in order to selectively study only a few atomic layers close to the surface.

All these experiments are typically performed at low magnetic fields, where the Zeeman interaction for the electronic as well as the nuclear angular momentum is small compared to the hyperfine interaction between them. Accordingly, the relevant optical properties of the system cannot be related to either of these spins individually, but to the total angular momentum

$$\boldsymbol{F} = \boldsymbol{J} + \boldsymbol{I} = \boldsymbol{L} + \boldsymbol{S} + \boldsymbol{I},$$

where $\boldsymbol{I}$ represents the nuclear spin, $\boldsymbol{S}$ the electron spin, $\boldsymbol{L}$ the orbital angular momentum of the electron and $\boldsymbol{F}$ the total angular momentum of electron and nucleus. Only when the electronic Zeeman interaction is stronger than the hyperfine interaction can the two angular momenta be considered independently.

## 3.2 Detection of nuclear spins

Since the nuclear spin does not couple to the electric dipole moment of optical transitions, direct optical detection of NMR is not as straightforward as for electron spins. However, since nuclear spins interact with electron spins, indirect detection schemes are possible and have been explored. Here, we can only discuss a few of the possible schemes and we focus on NMR transitions in electronic ground states, which is probably the most relevant situation.

We first consider the case where the electronic ground state has unpaired electrons, i.e. $S > 0$. In this case, the hyperfine interaction is typically the dominant interaction for the nuclear spin. It is then strongly coupled to the electron angular momentum, as discussed at the end of section 3.1 and can be detected exactly in the same way as the electron spin.

The situation is less straightforward in systems where the electronic angular momentum vanishes, $J = 0$. In this case, $\boldsymbol{F} = \boldsymbol{I}$ and the interaction with magnetic fields is the same as in other diamagnetic systems. However, the transition matrix elements of optical transitions still depend on the state of the nuclear spin. Fig. 11 shows the basic principle for a transition between the electronic ground state with $L = S = 0$ and an electronically excited state with $L = 1, S = 0$. In this case, angular momentum





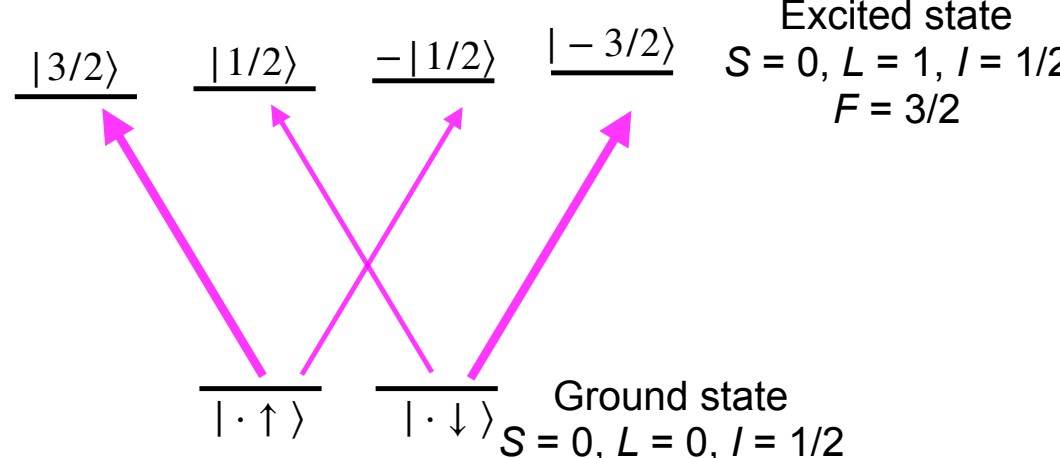

**Figure 11.** Detection of a nuclear spin state in a system with vanishing electron spin in the ground state. The transition strength of the thinner arrows is $\sqrt{3}/3 \approx 0.58$ times that of the thicker ones.

selection rules imply that only the transitions indicated by arrows in Fig. 11 have non-vanishing transition moments for light propagating along the quantisation axis. The transition strength of the thinner arrows is $\sqrt{3}/3 \approx 0.58$ times that of the thicker ones. In addition, the excited states interact with a magnetic field much more strongly than the nuclear spin and therefore they split in a magnetic field, so these allowed transitions are non-degenerate. Accordingly, the nuclear spin state has a significant influence on the transition probabilities, which allows one to determine the spin state from either absorption or emission measurements (Takei et al., 2010). Enhancing the interaction between the radiation field and the atomic system, e.g. by a resonant cavity, even allows one to detect single nuclear spins in such a system (Takei et al., 2010).

### 3.3 Detection of NMR by optical magnetometry

Light transmitted through atomic vapors provides a very sensitive detector for magnetic fields, as discussed in sections 2.4.1 and 3.1. Accordginly, this type of measurement can be used to detect magnetisation from nuclear spins - either inside the optically active medium or outside. The first case (nuclear spins in the active medium) is the typical situation for magnetic resonance of noble gases, in particular $^3$He and $^{129}$Xe.

Figure 12 shows the basic setup for optically detected NMR of noble gas atoms in an optical magnetometer. The pump laser beam generates spin polarisation of alkali atoms, typically Rubidium (Rb). During collisions with the noble gas atoms (typically $^{129}$Xe or $^3$He), part of their spin polarisation is exchanged with the nuclear spins of the noble gas atoms, which results in significant nuclear spin polarisation. The nuclear as well as the electronic spins evolve in the total magnetic field, which includes contributions from an external bias field (which may be zero), the magnetization of the Rb atoms and the nuclear spin magnetization of the noble gas atoms. The resulting polarization of the Rb atoms is detected through their effect on the polarisation of the transmitted probe beam. A change in any component of the total magnetic field is therefore observed





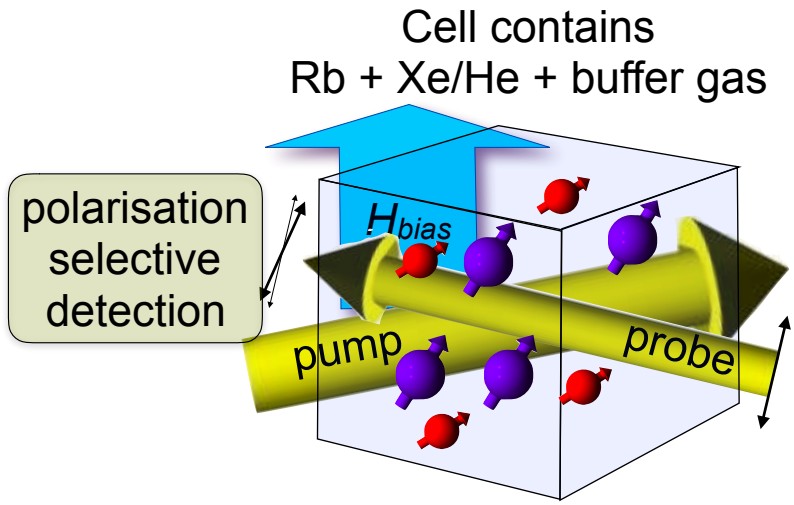

**Figure 12.** Setup for zero- and low-field NMR in an optical magnetometer based on alkali atomic vapor.

as a change of the signal from the polarisation selective detector. This allows optical detection also of the nuclear spins of the
rare gas atoms (Savukov and Romalis, 2005; Ledbetter et al., 2009, 2011).

Since the nuclear spins in this system do not interact directly with the laser beam, they can be spatially separated from the
Rb magnetomter to allow detection of spin species that are not compatible with an atomic vapour (Budker and Romalis, 2007;
Ledbetter et al., 2011). The detection sensitivity of atomic-vapor based magnetometers is given by the magnetic flux, rather
than its derivative with respect to time (as in inductive detectors). Accordingly, it does not increase with the Larmor frequency
and is often higher at low frequencies and thus low fields. This makes it a particularly attractive tool for measuring NMR
spectra in "ultra-low fields", where the Zeeman effect is a small perturbation of the zero-field Hamiltonian, which is usually
dominated by $J$-couplings for liquid-state NMR (Appelt et al., 2010).

The Hanle-effect discussed in more detail in section 5.2 for semiconductors uses a closely related effect: in these systems,
the spin of the electrons in the excited states undergo Larmor precession in a magnetic field that is given by the sum of the
externally applied magnetic field and an effective field due to the average interaction with a large number of nuclear spins,
which is known as the nuclear field.

## 4 Dielectric solids

### 4.1 Rare-earth ions

Dielectric crystals hosting rare earth ions (REIs) have been studied by optical spectroscopy for many years. The main motiva-
tion for these studies derives from the relatively narrow transitions between different electronic states which differ mostly with
respect to the configurations of their $f$-electrons. The transitions between these states are therefore "forbidden", i.e. they have





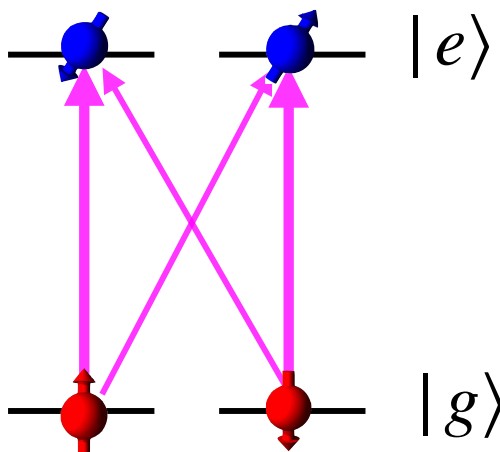

**Figure 13.** Eigenstates of the nuclear spin in the ground- and electronically excited states of a rare-earth ion. The quantization axes (indicated by the direction of the arrows) of ground- and excited state have different orientations, not along the magnetic field. The pink arrows between the ground- and excited states indicate optical transitions with the widths representing different transition dipole moments.

small transition dipole moments and long lifetimes. Furthermore, the electrons are relatively well shielded from perturbations by charged defects, resulting also in a relatively small inhomogeneous broadening of the transitions.

The nuclei of many rare-earth isotopes have nonzero spin, which interact with external magnetic fields as well as with the
angular momenta of the electronic system. One of the consequences is that the nuclear spin eigenstates of different electronic states are not identical. Figure 13 shows the situation for a nuclear spin $I = 1/2$. Due to the hyperfine interaction, the quantisation axes of the nuclear spin in the ground- and electronically excited states have different orientation. In the figure, the quantisation axes are marked schematically by the orientation of the spins. As the system undergoes electronic excitation, the new state has overlap with both nuclear spin states and accordingly the transition dipole moments for all four possible
transitions are non-zero.

This is a very important precondition for the optical excitation and detection of NMR transitions by coherent Raman scattering (CRS). As discussed in section 2.4.3, Raman excitation requires that two optical transitions sharing one energy level have non-vanishing transition dipole moments. According to figure 13, this is fulfilled for V-type as well as Λ-type transitions in REI systems, which allows one to use CRS for studying ground- as well as excited states.

Figure 14 shows two examples of NMR spectra of $^{141}$Pr, a typical rare-earth isotope with a nuclear spin of 5/2. For these experiments, it is added as a dopant to the host crystal YAlO$_3$ (Klieber and Suter, 2005). The upper spectrum shows the NMR transitions when the ion is in the electronically excited $^1D_2$ state, while the lower spectrum shows the same transitions for the $^3H_4$ electronic ground state. The large differences in the transition frequencies originate from the different hyperfine interactions, which depend on the electronic configuration.





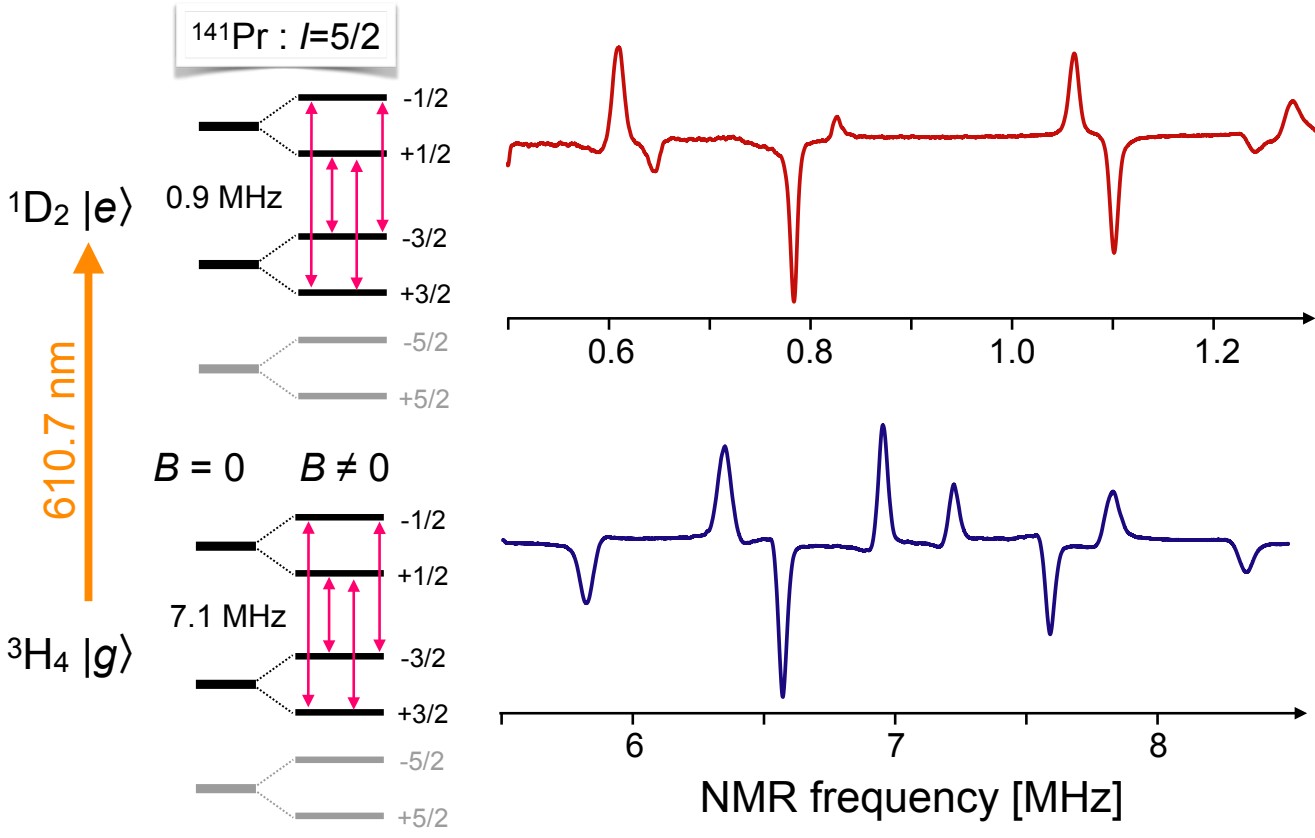

**Figure 14.** Energy levels and NMR spectra of $^{141}$Pr in the ground- and electronically excited states of Pr$^{3+}$ in an YAlO$_3$ host crystal, measured by coherent Raman scattering. The grey parts of the level scheme do not contribute to the spectra shown in the figure but are included for completeness. The four positive lines correspond to the transitions of one crystallographic site, as marked in the level scheme. The four negative lines belong to the corresponding transitions in a different crystallographic site.

One of the most exciting applications of optically excited and detected magnetic resonance in rare-earth ions is the possibility to use these materials as memories for quantum states. This is an important prerequisite for many emerging quantum technologies, such as quantum communication. Quantum states can be stored in the relatively long-lived electronic states of the rare-earth ions, but transferring them into nuclear spin degrees of freedom can extend the lifetime by several orders of magnitude to the range of seconds (Lovrić et al., 2013) and, for custom-designed crystals, even to several hours (Zhong et al.,

2015) - by far the longest lifetime of a quantum memory measured so far.

The experimental techniques for studying these materials include transmission as well as fluorescence experiments, CW as well as time-resolved experiments. Pioneering work on these systems was performed, e.g. in the group of Brewer, which demonstrated Raman-heterodyne detection, pulsed as well as CW on $^{141}$Pr ions substitutionally doped into a LaF$_3$ crystal (Mlynek et al., 1983). Excitation of the spin transitions can be performed by RF fields as well as purely optically, e.g. with a




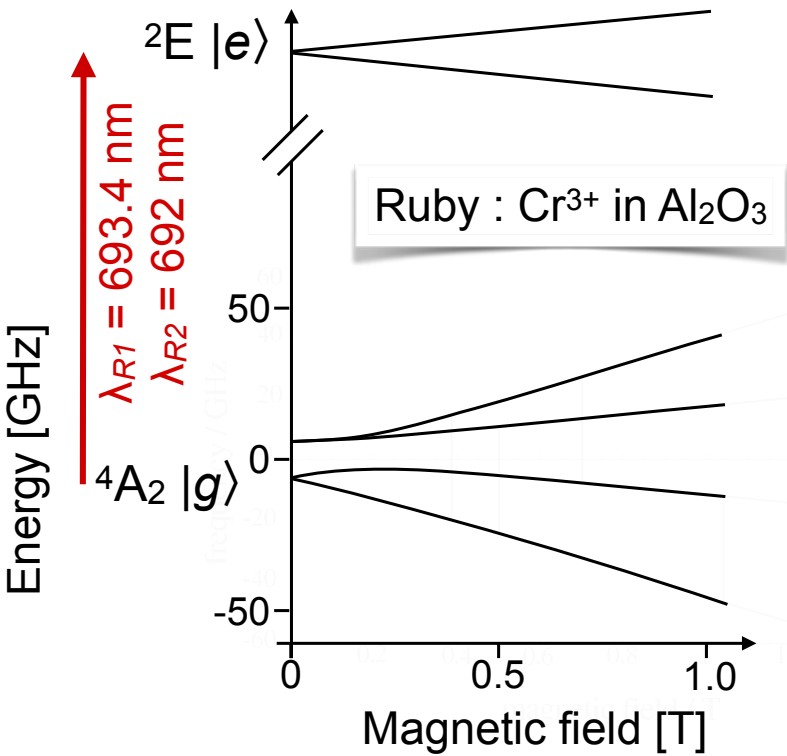

**Figure 15.** Partial energy level scheme of $Cr^{3+}$ in Ruby.

bichromatic laser field that excites the spin transition via a coherent Raman process (Blasberg and Suter, 1994, 1995). The two
frequency components of the laser field should then be separated by the transition frequency of the spins.

## 4.2   Transition metal ions

Transition metal ions have optical resonance lines that are much broader then rare earth ions and depend much more on the
environment of the ions. This makes their use in optical detection more challenging but also potentially more rewarding.
Transition metal ions are essential components of many biologically relevant molecules,

The most popular test system in this category is certainly Ruby, i.e. $Cr^{3+}$:$AlO_3$ (Geschwind et al., 1965; Börger et al., 1999).
Figure 15 shows the energy level system of the electronic ground state and one of the excited states. The transition between
the ground state and the $^2E$ excited state is known as the $R_1$ transition (or $R_2$ transition for a nearby state). Using Ruby
as a testbed, Geschwind et al. (Geschwind et al., 1965) explored several methods for measuring magnetic resonance in the
electronic ground- as well as in excited states. Their approach compared the measurement of circular polarisation, selective
reabsorption and high-resolution spectrometry. The same system was also studied by purely optical methods like photon-echo
modulation (Szabo, 1986).





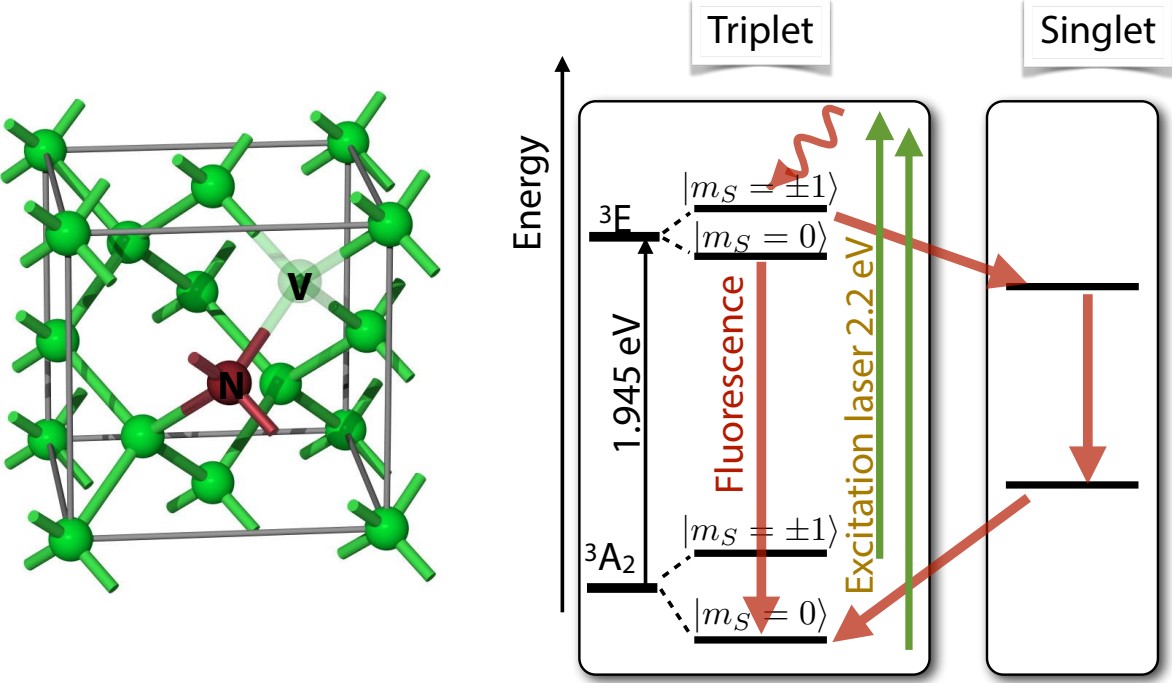

**Figure 16.** Structure of the nitrogen-vacancy (NV) defect center in diamond and energy level system. Optical excitation results in spin-dependent intersystem crossing (ISC) to the singlet state, which occurs predominantly for the $m_S = \pm 1$ states and therefore results in preferential population of the $m_S = 0$ state of the $^3A_2$ ground state.

More challenging but also more interesting in terms of the potential information are measurements on biological macro-molecules like metalloproteins. In these systems, the optical detection approach provides more information than classical inductive detection, since it relates the magnetic anisotropy to the electronic and molecular structure of the molecule (Börger et al., 1999; Bingham et al., 2000a, b; Börger et al., 2001; Schweika-Kresimon et al., 2002). While the analysis of these correlations is not trivial, it allows one to obtain detailed information on the electronic and atomic structure of the molecules in frozen solutions, sometimes circumventing the need for growing single crystals.

### 4.3 The NV-center of diamond

One of the most popular systems for testing magnetic resonance with single spins is the nitrogen-vacancy (NV) center in diamond (Manson et al., 2006; Doherty et al., 2013; Suter and Jelezko, 2017). Figure 16 shows on the left the structure of the center: one carbon of the diamond lattice is replaced by a nitrogen atom, while one of its nearest neighbours is missing: the corresponding atom is replaced by a vacancy. The center is most useful in the negatively charged state (NV$^-$). It contains then a total of 6 electrons : 3 from the dangling bonds next to the vacancy, 2 from the lone pair at the nitrogen and the single additional electron that has been captured from one of the available donors. The $^3A_2$ ground state of these 6 electrons has spin

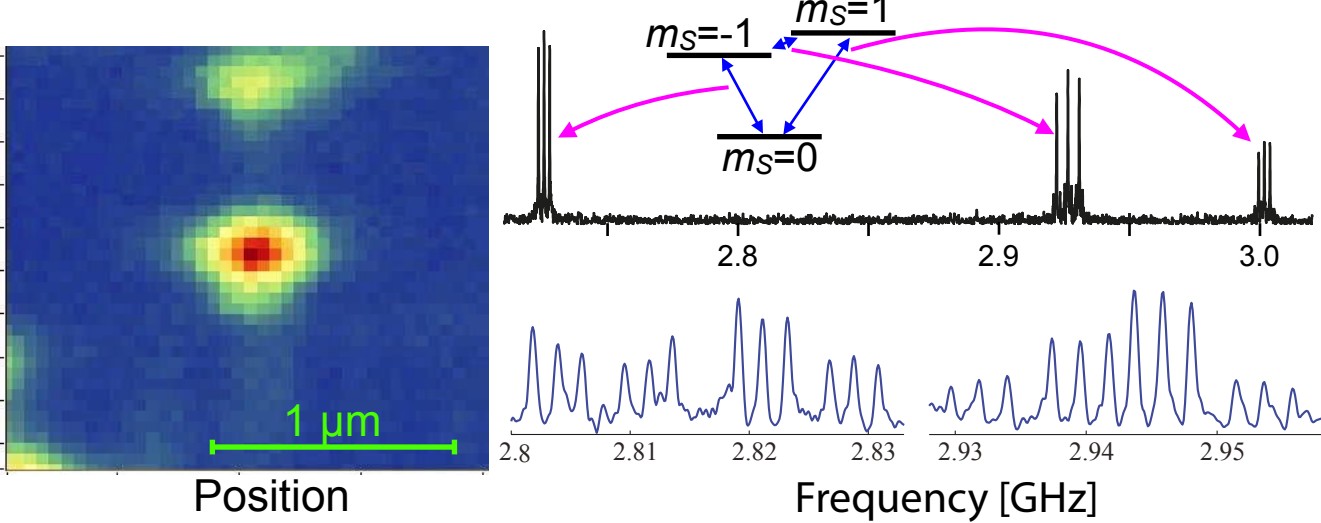

**Figure 17.** Left: scanning confocal image of a single NV center. Right: Optically detected ESR spectra of 2 different NV centers showing hyperfine interaction with the $^{14}$N (top) and the $^{14}$N plus a $^{13}$C (bottom) nuclear spin.

$S=1$, and it can be optically excited into a $^3$E state. This transition does not normally change the spin state and in most cases, the center falls back into the same ground state. However, there is a small probability for the center to undergo inter-system crossing (ISC) into the singlet manifold. This probability is significantly larger for centers in the $m_S = \pm 1$ state than in the $m_S = 0$ state. From the singlet manifold, the system falls back into the ground state but during this process, the spin state becomes randomised. The overall effect is that the $m_S = 0$ state becomes significantly more populated than the other states. This process does not require polarised light and can be easily observed in conventional ESR experiments, where irradiation with white light results in a strongly enhanced ESR spectrum (Loubser and van Wyk, 1978).

When the system takes the "detour" through the singlet state, this requires significantly more time than the direct path back to the ground state, and it does not emit a photon during this cycle. Accordingly, the centers generate some 20 % fewer photons when they are initially in the $m_S = \pm 1$ state. Counting the number of photons emitted per unit time is therefore a simple way of detecting the spin state of the system at the start of the counting period. This readout is destructive, as the system is forced into the $m_S = 0$ state by the readout process, but it can be used to detect, e.g., magnetic resonance spectra (Doherty et al., 2013; Suter and Jelezko, 2017).

Figure 17 shows two examples of ESR spectra. The upper one shows three transitions, two of them corresponding to "allowed" $\Delta m_S = \pm 1$ transitions and the central one to the "forbidden" $m_S = -1 \leftrightarrow m_S = +1$ transition (Niemeyer et al., 2013). All three transitions are split by the hyperfine interaction with the $^{14}$N ($I = 1$) nuclear spin. For the $\Delta m_S = \pm 2$ transition, the splitting is twice as large as for the $\Delta m_S = \pm 1$ transitions.

The lower trace shows the ESR spectrum from a different center, where one of the neighbouring atoms is a $^{13}$C nuclear spin, which also has a hyperfine interaction with the electron. While the interaction with a $I = 1/2$ nuclear spin normally leads to a




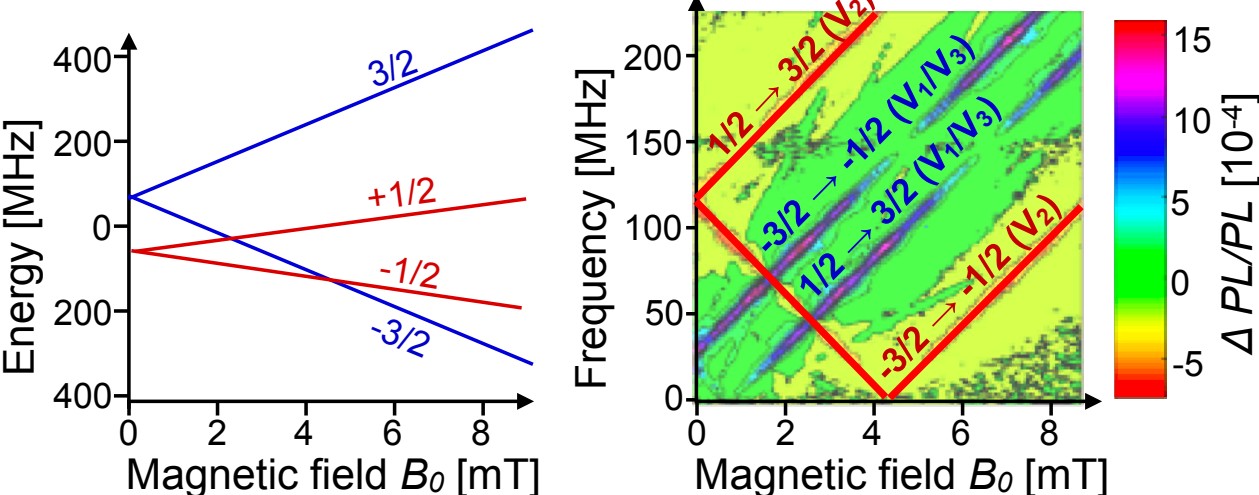

**Figure 18.** Left: Energy level system of the $V_2$ Si$^-$ vacancy in SiC (6H polytype). Right: Experimental ODMR spectra as a function of the magnetic field (horizontal axis) and RF frequency (vertical axis). The experimental spectrum contains transitions from $V_1$ and $V_3$ vacancies. The color indicates the relative change in PL.

splitting into a doublet (of the triplet due to the $^{14}$N interaction), it results here into a quartet (Rao and Suter, 2016). This is a consequence of the different orientations of the nuclear spin quantisation axes in the different electron spin states. As shown in figure 13, this implies that all four possible transitions become (partly) allowed.

The hyperfine interaction allows one not only to observe EPR transitions, but also nuclear spin transitions - in many cases even without applying RF pulses (Zhang et al., 2019). While the spectra shown here are all associated with magnetic resonance

of electronic and nuclear spins that are part of the NV center, it is also possible to use NV centers as sensors for indirectly detecting more remote spins, either electronic or nuclear ones (Aslam et al., 2017; Glenn et al., 2018).

### 4.4   Silicon-carbide

Silicon-carbide is a material that is closely related to diamond: its structure can be derived from diamond by replacing alternating carbon sites with silicon. It also shares other properties like large bandgap and high mechanical strength. There are also

major differences, in particular there is not just a single structure, but the material has a large number of polytypes, where successive SiC layers follow different stacking patterns. In each polytype, several active spin-centers have been described (see, e.g., Falk et al. (2013)), mostly silicon and carbon vacancies, as well as divacancies. Depending on their charge state, they have a spin of 1/2, 1 or 3/2 (in units of $\hbar$).

Figure 18 shows on the left-hand side the energy level scheme of a typical spin center in SiC: A Si$^-$ vacancy in the 6H

polytype. Depending on the site, there are several such centers - the one shown here is the $V_2$ center, which has a zero-field splitting of 128 MHz between the $m_S = \pm 1/2$ and the $m_S = \pm 3/2$ states (Biktagirov et al., 2018). The right-hand part of the





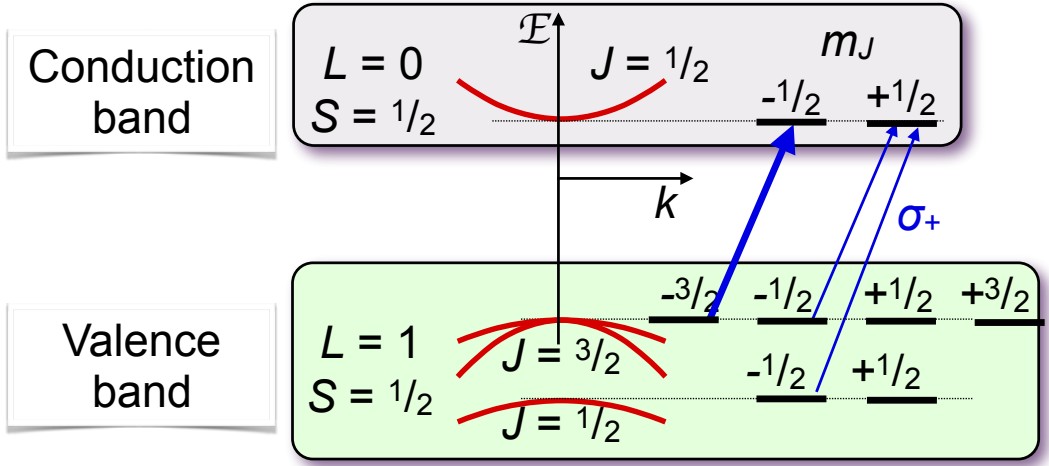

**Figure 19.** Optical pumping in a direct-band-gap semiconductor. The arrows mark the allowed optical transitions for circularly polarised light.

figure shows the ODMR spectrum, measured by the change in photoluminescence when an RF field is applied, as a function of the magnetic field and the RF frequency. The change of the PL intensity is color-coded according to the scale bar on the right. The experimental data contain signals from the $V_2$ centers shown in the left, as well as from the $V_1/V_3$ centers, whose zero-field splitting is -28 MHz. The ODMR signal from the $V_1/V_3$ centers is positive (i.e. the PL increases when an RF field is applied), while the signal from the $V_2$ centers is negative (Singh et al., 2020).

## 5 Semiconductors

### 5.1 Optical pumping

Optical pumping, i.e. the transfer of angular momentum from photons to electronic and nuclear spins, can follow two distinct paths in semiconductors, depending if the photons are absorbed by localized defect states such as deep donors or if they target directly the delocalized electrons, raising them from the valence- to the conduction band. A good example for the work with localized defects is Ref. (Koschnick et al., 1996), which used the 2.2 eV transition of a residual donor in nominally undoped GaN to measure the nuclear spin transitions of both Ga isotopes via optically detected ENDOR.

Figure 19 shows the basic principle of optical pumping via delocalized electrons for a semiconductor with a direct band-gap, such as GaAs, where the bottom of the conduction band is at the same linear momentum as the top of the valence band. These materials are particularly useful for light-emitting devices, such as lasers and LEDs. The electronic states in the valence band, which has a $p$-type character, have total angular momentum $J = 3/2$, while the states in the conduction band ($s$-character) have angular momentum $J' = 1/2$. A circularly polarised laser field therefore couples only the transitions indicated in the figure (or



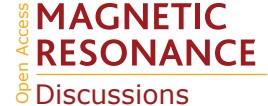

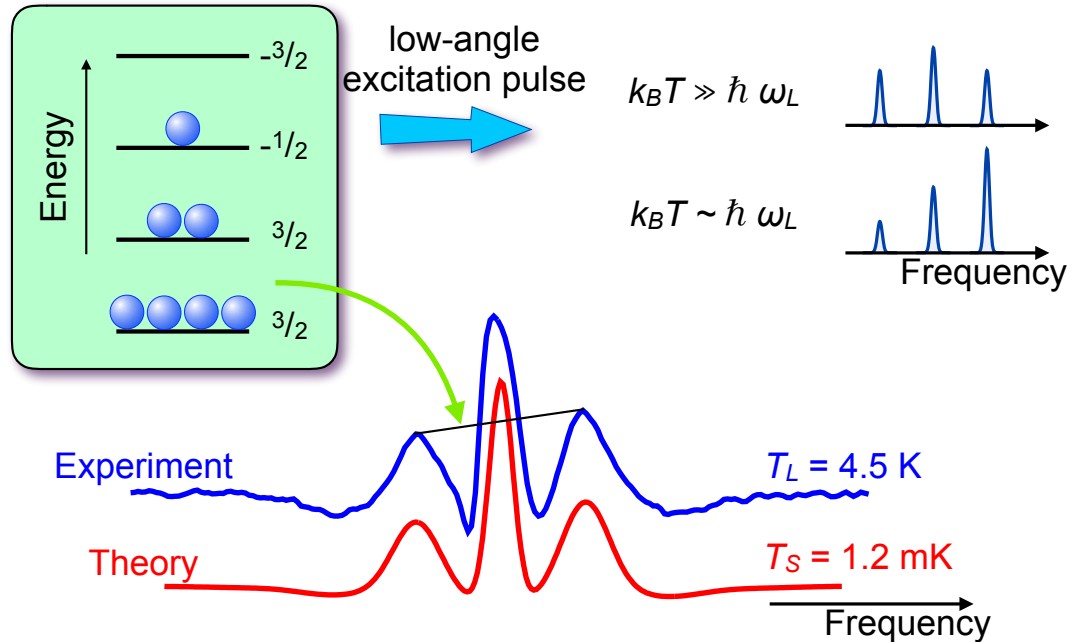

**Figure 20.** Experimental and simulated NMR spectra of $^{75}$As in GaAs, measured with a small flip angle. The different amplitudes of the quadrupole satellites are then proportional to the population differences across the corresponding transitions and therefore allow a direct measurement of the spin temperature.

the corresponding mirror image for opposite circular polarisation). Optical excitation is most efficient if the photon energy is

close to the exciton energy, thereby generating charge carriers in both bands with minimal kinetic energy.

If pumping is done in a magnetic field, the eigenstates of the electron system are the Landau levels and optical pumping above the band gap must be described in terms of Landau levels (Mui et al., 2009; Wheeler et al., 2016).

The mechanism shown in fig. 19 requires photon energies at or above the bandgap. While this works for all systems, it is also possible to use below-band-gap light for pumping. In this case, the photons are absorbed by localised centers, such as donors,

whose energy levels are inside the band-gap (Pietrass et al., 1996). For some samples, this process is actually more useful, particularly in bulk material, since the absorption coefficient is smaller. The light can then penetrate farther into the sample and achieve more homogeneous pumping.

The excitation process directly pumps only the electronic spins. From there, the spin polarisation also flows to the nuclear spins via the hyperfine interaction, as discussed in more detail in section 5.3. The efficiency of this process depends strongly

on the system, but in some cases, it can cool the nuclear spins to very low temperatures, as in the example shown in Figure 20. Here the spectrum of $^{75}$As, which has a spin $I = 3/2$ in GaAs is split by the quadrupole interaction, which allows one to observe all three transitions independently. If the system is excited by an RF pulse with a small ($\ll \pi/2$) flip angle, the amplitudes of the lines are proportional to the population differences, which allows one to measure the spin temperature of the



system. In the example shown here, the spin temperature is $T_S \approx 1.2$ mK, while the sample (lattice) is at a temperature of $T_L \approx$ 4.5 K.

## 5.2 Hanle effect

If the pumping occurs at or above the band gap, it generates an electron-hole pair. When they recombine, the emitted photon carries again angular momentum. In the absence of relaxation and magnetic fields, angular momentum is conserved and the emitted photon has the same polarization as the photon that was absorbed. Under typical optical pumping conditions, this is circular polarisation. If relaxation is taken into account, the spin polarization and therefore also the polarization of the photoluminescence (PL) is reduced. If, in addition, a magnetic field is present that is not collinear with the excitation laser, it causes Larmor precession of the photo-induced spins (electrons and holes), which results in reduced PL polarisation in the direction of excitation but may generate polarization in a different direction. We now summarise these processes and discuss them in the so-called Hanle effect (Hanle, 1924; Ellett, 1924; Wood and Ellett, 1924; Kastler, 1946; Paget et al., 1977).

Under steady-state illumination, the charge carriers in both bands settle into a stationary state that is determined by the interplay between the generation of new spins (along the direction of propagation of the incident laser), relaxation (reducing the polarization and the number of charge carriers) and Larmor precession in the magnetic field. For the conduction band electrons, these contributions can be summarized by an equation of motion

$$\frac{d\rho}{dt} = -\frac{i}{\hbar}[\mathcal{H},\rho] - \Gamma_r\rho - \hat{\Gamma}_s + \hat{P}', \tag{2}$$

where we have assumed for simplicity that the optical pumping populates only one of the spin states. This is a good approximation for the case of quantum wells, where the transitions between the $m_J = -3/2 \rightarrow m_{J'} = -1/2$ and $m_J = -1/2 \rightarrow m_{J'} = 1/2$ are not degenerate in zero magnetic field. In eq. (2), $\Gamma_r$ is the radiative decay rate and the spin relaxation tensor $\hat{\Gamma}_s$, the Hamiltonian $\mathcal{H}$ and pumping matrix $\hat{P}'$ are

$$\hat{\Gamma}_s = \Gamma_1 \begin{pmatrix} \frac{\rho_{11}-\rho_{22}}{2} & \rho_{12} \\ \rho_{21} & -\frac{\rho_{11}-\rho_{22}}{2} \end{pmatrix}$$

$$\mathcal{H} = \hbar\gamma_e \boldsymbol{B}\cdot\boldsymbol{S}$$

$$\hat{P}' = \begin{pmatrix} P & 0 \\ 0 & 0 \end{pmatrix}.$$

Here, the direction of the laser beam was chosen to be along the $z$-axis, so the absorption of a photon creates electron spins along the $+z$ axis. $P$ is the rate at which the laser beam generates electron spin density in the conduction band and we have assumed that the spin relaxation rate $\Gamma_1$ is the same for all three components of the spin, $\Gamma_1 = 1/T_1 = 1/T_2$.

To solve the equation of motion, it is often more convenient to choose the $z$-axis along the magnetic field direction. Writing $\theta$ for the angle between the directions of the magnetic field and the laser beam, the pumping matrix is then

$$\hat{P} = \frac{P}{2} \begin{pmatrix} 1+\cos\theta & \sin\theta \\ \sin\theta & 1-\cos\theta \end{pmatrix}.$$





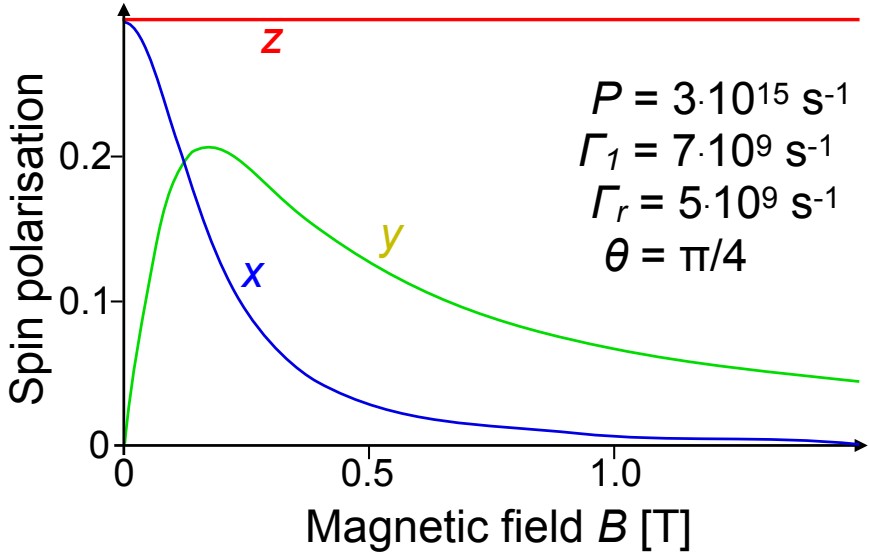

**Figure 21.** Polarization of the components of the electron spin in the conduction band as a function of the magnetic field $B$ applied along the $z$-axis. The laser is incident in the $xz$-plane, at an angle of $45°$ from the magnetic field. The remaining parameters are shown in the figure.

Solving this for the stationary condition, one obtains the steady-state electron density in the conduction band as the trace of the density operator,

$$\rho_{11} + \rho_{22} = \frac{P}{\Gamma_r}$$

and the $x, y$ and $z-$ components of the polarisation vector of the electron spin are

$$
\begin{aligned}
\rho_{12} + \rho_{21} &= P\frac{\sin\theta(\Gamma_r + \Gamma_1)}{(\Gamma_r + \Gamma_1)^2 + \Omega_L^2} \\
-i(\rho_{12} - \rho_{21}) &= P\frac{\Omega_L}{(\Gamma_r + \Gamma_1)^2 + \Omega_L^2} \\
\rho_{11} - \rho_{22} &= \frac{P\cos\theta}{\Gamma_1 + \Gamma_r}.
\end{aligned}
\tag{3}
$$

Since the polarization of the photoemission depends on the spin polarisation of the electrons in the conduction band (and in general also on the polarization of the holes in the valence band), this also determines the photoluminescence (Hanle, 1924). Figure 21 shows the dependence of the spin polarization when the laser beam is incident in the $xz$-plane and the magnetic field is applied along the $z$-axis. While the $z$-component is not affected by the magnetic field, the transverse components show a typical absorption / dispersion behavior. The width of the line is given by the electron spin relaxation rate $\Gamma_1$ and the radiative

lifetime $\Gamma_r$, which can be measured indpendently (Schreiner et al., 1992). This was first discussed by Hanle in the context of atomic fluorescence and observed in semiconductors by Parsons (Parsons, 1969). It is still used in the resonance fluorescence from atomic vapors, for the detection of magnetic fields or nuclear spins in zero and ultra-low fields, as discussed in section 3.3.





### 5.3 Dynamic nuclear polarisation

The hyperfine interaction can exchange polarisation between electronic and nuclear spin, resulting in significant polarisation of the nuclear spins. Depending on the system, this process can involve different steps. Here, we first discuss a process where the optical irradiation generates electron spin polarization on a localized center, such as a shallow donor in GaAs (Paget, 1982) or an NV-center in diamond (Doherty et al., 2013; Suter and Jelezko, 2017). A typical experiment on diamond starts with a laser pulse that initializes the electron spin into the $m_S = 0$ ground state. The electron spin is coupled to the $^{14}$N nuclear spin

of the NV center as well as to many $^{13}$C nuclear spins at random locations in the lattice. The hyperfine interaction with the $^{13}$C nuclear spins reaches a maximum value of $\approx 130$ MHz (Rao and Suter, 2016) for nuclei located next to the vacancy. For more remote nuclei, it decreases with the third power of the distance. Depending on the interaction strength, the directly coupled nuclear spins also acquire some polarisation from the electron spin, but this process can be made much more efficient by the application of microwave fields (Ajoy et al., 2018b; Zangara et al., 2019). From the directly coupled nuclear spins, the

polarisation spreads to the bulk of the crystal through nuclear spin diffusion.

If the charge carriers are delocalised, their polarization can also be transferred to nuclear spins through the Overhauser effect (Lampel, 1968) or by dynamic nuclear polarization driven by microwave irradiation. The rate at which the nuclear spin polarisation builds up is close to the spin-lattice relaxation rate. These rates can vary significantly over different materials and doping levels but are typically of the order of minutes at low temperature (Schreiner et al., 1997).

### 5.4 Nuclear field

In the most important semiconductors, the conduction band is formed by $s$-type orbitals. The electronic spins in the conduction band therefore interact with the nuclear spins in the material through hyperfine interactions dominated by the Fermi-contact term. If the electrons are localised (e.g. near interface fluctuations or impurities such as shallow donors), their wavefunction has a radius of the order of tens of nm, which means that at least $10^5$ nuclear spins have direct contact with the electron spin.

The interaction strength with a single nuclear spin is of the order of some tens of kHz. From the point of view of the electron, this is very small compared to the other interactions. As long as the nuclear spins have thermal polarisation, i.e. roughly equal populations of $\uparrow$ and $\downarrow$, their effect on the dynamics of the electron spin is therefore small. However, since the number of nuclear spins that interact with the electron is very large, these small contributions can add up to very large effects. The first is a dephasing effect: due to statistical fluctuations ("spin noise"), the electron interacts with a fluctuating environment that can

result in dephasing of the electron spin.

If the nuclear spins are polarised ($\rightarrow$ ch. 5.3), their interactions do not cancel but add up and their combined effect can be very significant and become the dominant interaction for the electrons. For large polarisation, the effect is comparable to a magnetic field of the order of several Teslas (Paget et al., 1977; Chekhovich et al., 2017). The effective nuclear field exists for localised as well as for mobile electrons, but the effect differs. For metals, the effect was calculated by Overhauser (Overhauser,

1953). Figure 22 shows the build-up of the nuclear field in GaAs during optical pumping. This effective magnetic field adds to the external field and the evolution of the electron spins is determined by their sum. If one measures the polarization of the





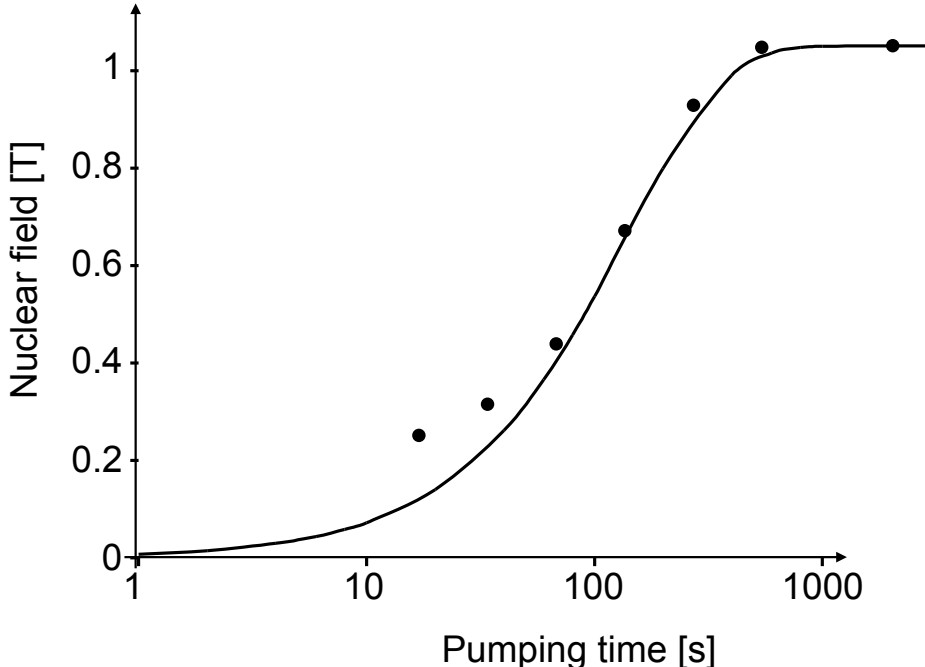

**Figure 22.** Build-up of the nuclear field during optical pumping.

photoluminescence as a function of a transverse magnetic field, one therefore observes a shifted Hanle curve whose maximum marks the external field that has the same magnitude but opposite sign as the internal nuclear field.

### 5.5 Detection of NMR via Hanle-effect

Figure 23 shows the emission of a photon during recombination. The energy level scheme shown here is that of a quantum well (a very thin layer of GaAs sandwiched between barriers of AlAs), and in the valence band, only the $J = 3/2$ levels are shown. Compared to the level scheme of figure 19, which represents the levels of a bulk material, the $J = 3/2$ levels are not degenerate in the quantum well. The situation shown in the figure corresponds to a single electron in the conduction band, in the $J_z = -1/2$ state. In the valence band, a single hole exists in the $J_z = -3/2$ state, e.g. because it was created there by

the absorption of a photon. If the electron and hole recombine to emit a photon, their angular momentum must be carried away by the photon, which is circularly polarised. Accordingly, the spin polarisation of the emitted photon directly reflects the polarisation of the electron spin, which can be modified by relaxation and by magnetic fields, as discussed in section 5.2. When polarised nuclear spins are present, the nuclear field adds to the external field and therefore shifts the Hanle curve: The maximum of the PL polarisation is no longer at zero external field, but at an external field that is exactly equal in magnitude

but opposite in direction to the nuclear field. The sum of the external and the nuclear field is thus zero and the electron spin polarisation is not degraded (Dyakonov et al., 1975; Paget et al., 1977).

MAGNETIC
RESONANCE
Open Access Discussions

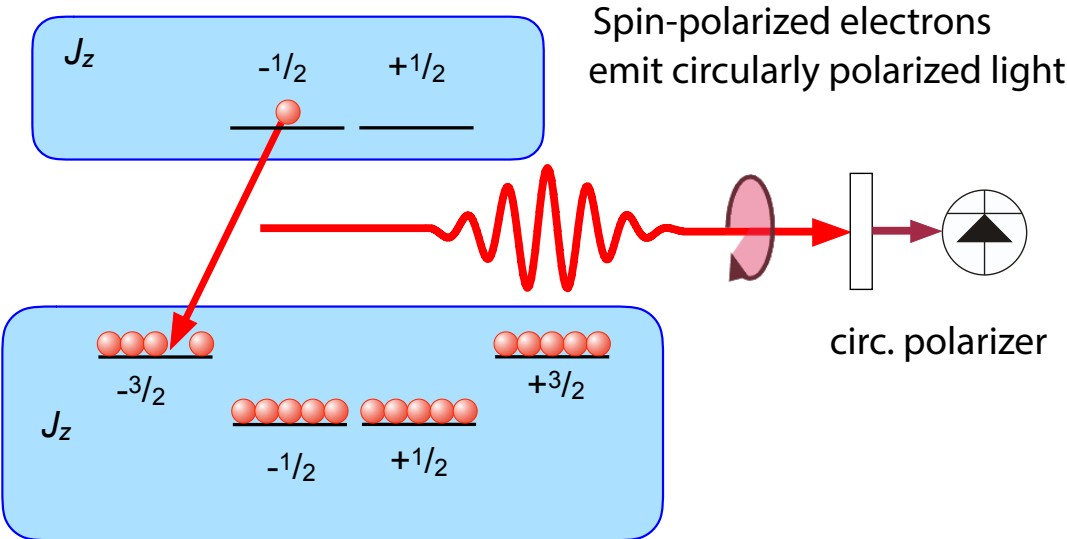

**Figure 23.** Photoluminescence (PL) from a semiconductor quantum well when the charge carriers are partly spin-polarised. For the case shown here, the emission perpendicular to the well is circularly polarised.

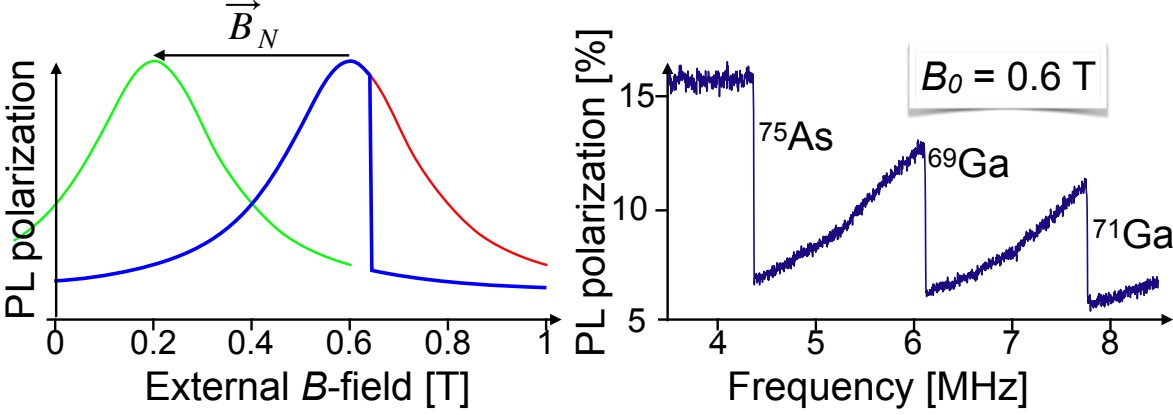

**Figure 24.** Principle of detection of NMR through the Hanle effect (left) and experimental NMR spectrum of a GaAs quantum well (right). PL = photoluminescence.

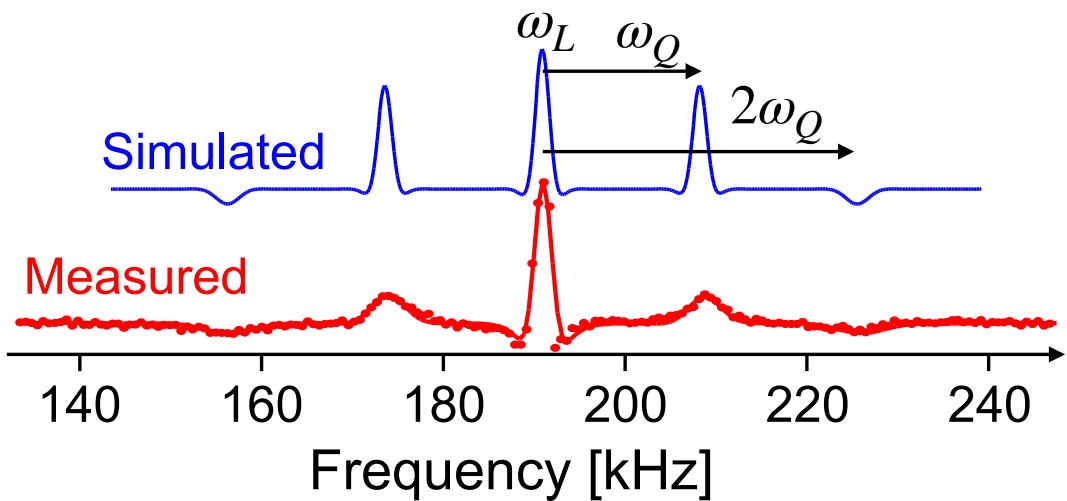

**Figure 25.** Spectrum of $^{75}$As obtained by Fourier transformation of the FID signal measured via the Hanle effect after pulsed excitation.

This effect can be used for detecting NMR transitions, as demonstrated in the 1970s and 1980s (Dyakonov et al., 1975; Paget et al., 1977; Paget, 1981, 1982) in GaAs. Figure 24 shows the basic principle of optical detection of NMR through the Hanle effect. The red curve represents the Hanle curve of a semiconductor system, whose nuclear field is of the order of 0.6 T. The

magnetic field is scanned upwards and at the same time a radio-frequency field is applied to the sample. At some 0.65 T, the resonance condition for one of the nuclear spin isotopes in the sample is fulfilled and the RF irradiation saturates the nuclear spins. Accordingly, the nuclear field drops to a much lower value and the PL polarisation drops to the value determined by the green curve, which represents the Hanle curve for a total nuclear field of only 0.2 T.

The right-hand part of the figure shows an experimental spectrum of GaAs, where the external magnetic field is kept constant

at 0.6 T, while a radio-frequency is applied whose frequency increases from 3.5 to 8.5 MHz. Whenever the frequency reaches the resonance frequency of one of the nuclear spin species present in the sample, it saturates the corresponding spin system, which leads to a reduction of the spin polarisation, the corresponding component of the nuclear field and therefore of the effective magnetic field. This is observed as a sudden drop of the PL polarisation at the positions where the resonance occurs for the three isotopes $^{69}$Ga, $^{71}$Ga, and $^{75}$As. Such experiments can target, e.g., individual quantum wells in a semiconductor

heterostructure (Eickhoff et al., 2003). They offer a number of applications, such as the measurement of the spin density at different sites within the unit cell from a measurements of the Knight shift (Krapf et al., 1991).

Similar to conventional NMR, spectra can be observed in CW mode, i.e., by observing the luminescence and scanning the RF in a constant magnetic field (Paget et al., 1977; Paget, 1981, 1982; Krapf et al., 1991), or in a time-resolved mode, by recording the freely precessing nuclear spin coherence that causes a modulation of the optical polarization (Eickhoff and Suter, 2004).

In this case, the nuclear field follows the Larmor precesion of the nuclear spins as determined by eq. (3) and the polarisation signal maps the Larmor precession. If the nuclear field is weak, the time-dependent polarisation signal corresponds directly to





the FID signal known from inductively detected NMR. If the nuclear field is strong enough, the transfer function from the spin polarisation to the observed signal is nonlinear.

Figure 25 shows, as an example, the Fourier transform of such an FID signal from $^{75}$As. The signals were obtained with a

nuclear field of approximately 1 T and two effects of the nonlinear response can be observed: The lineshape is distorted (with negative wings) and the spectrum contains not only the central transition and the two quadrupole satellites, but also additional resonance lines at $\omega_L \pm 2\omega_Q$, which result from mixing the centerband with the two sidebands by the nonlinear detector response. The simulated spectrum does not take quadrupolar broadening into account, which results in the larger width of the satellites in the experimental spectrum.

## 5.6   Other approaches to detection of NMR

The experiments discussed in section 5.5 use the effect of the nuclear spins on the polarisation of the electron spin and thus on the polarisation of the photoluminescence to detect the nuclear spin. This indirect detection scheme can be used in many similar experiments, such as optically detected ENDOR (Koschnick et al., 1996). In this experiment, optically detected EPR was performed by measuring the change in the total intensity of the photoluminescence when the microwave radiation was

turned on (Glaser et al., 1995). This signal was then further modulated by applying an additional RF field resonant with the nuclear spin transitions of nuclei that are coupled to the electron spin of the defect center. This experiment allowed a precise measurement of the hyperfine interaction between a donor electron and both Ga isotopes, which allowed a tentative assignment of the electron to an interstitial Ga.

Apart from these PL measurements, it is also possible to measure with transmitted (Teaney et al., 1960; Kikkawa and

Awschalom, 2000; Giri et al., 2013) or reflected (Stühler et al., 1994; Kikkawa et al., 1997) light. In both cases, the spin polarisation in the sample affects the complex index of refraction of the medium, which results in optical circular birefringence and optical circular dichroism, as discussed in section 2.4.1. If the light is off-resonant with respect to the optical transition, the effect is mostly through dispersion and is then known as Faraday effect.

## 5.7   Quantum films and quantum dots

Technologial developments in semiconductor science and technology often rely on so-called quantum-confined heterostructures, where the composition of the material varies over distances of a few nm. Ideally, these modifications use lattice-matched material systems, such as GaAs / AlAs, whose lattice constants differ by only $\approx 10^{-3}$. Accordingly, changing the composition does not significantly affect the structure, but changes the electronic properties. This allows one, e.g., to create effective potentials for charge carriers, an important prerequisite for many applications, such as lasers.

Studying these structures by the methods of magnetic resonance would be highly desirable, e.g., to assess the quality of the material and small variations of structural and electronic properties. Since structures with lateral dimensions of a few nm contain only a small number of spins, such studies can not be performed by magnetic resonance with direct detection. However, the large increase in sensitivity offered by optical techniques makes this relatively straightforward. Apart from providing



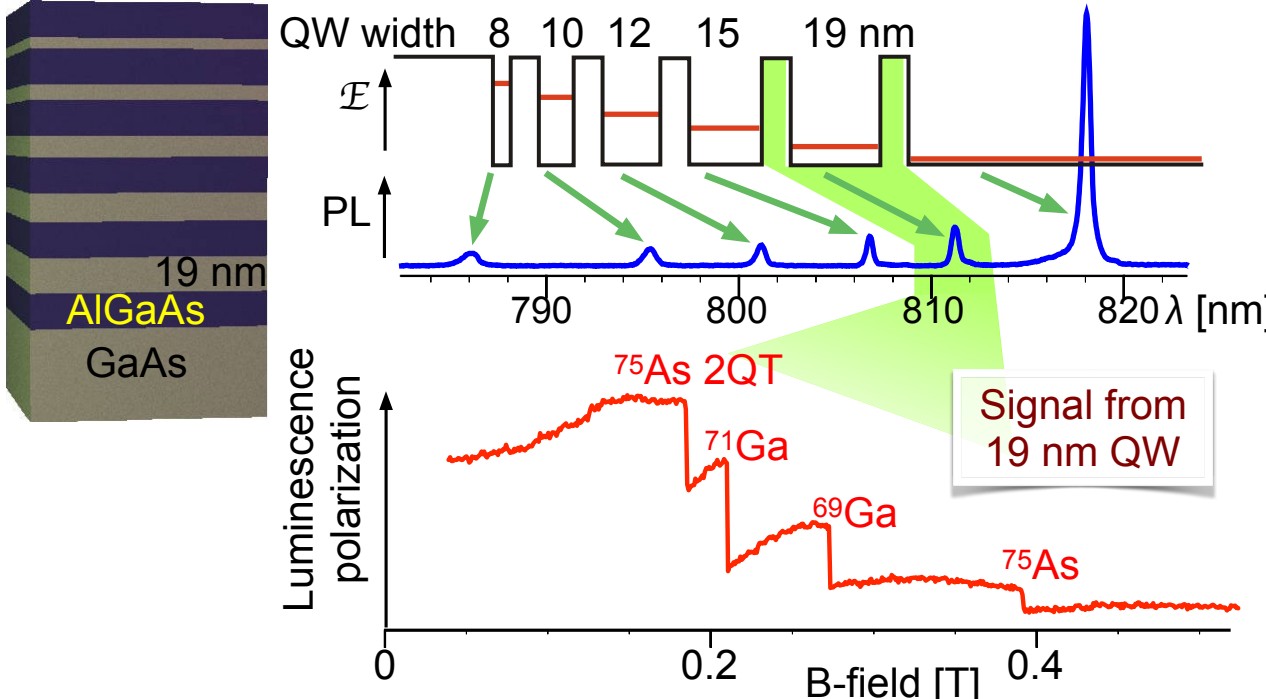

**Figure 26.** Optical detection of NMR in a single quantum well of a sample with five different quantum wells. Left: structure of the sample, where GaAs quantum wells of different thickness are separated by AlGaAs barriers. Top right: PL spectrum, with assignment of the individual resonance lines to specific quantum wells. Only light from the 19 nm quantum well is detected. Bottom right: NMR spectrum of the 19 nm quantum well detected by scanning the magnetic field while applying an RF field with a frequency of 5 MHz.

sufficient sensitivity, the additional degrees of freedom of the optical part of the experiment also allow one to selectively excite and detect only signal contributions arising from spins in a selected nanostructure.

Figure 26 shows how the wavelength of the photons can be used to select a specific part of the sample - in this case the 19 nm quantum well. In this example, the sample contains five different quantum wells with different thicknesses. Under non-resonant excitation, all quantum wells generate photons whose wavelengths are characteristic for the thickness of the corresponding quantum wells. In the shown example, the photons were separated in a monochromator and only the signal from the photons originating in the 19 nm quantum well was analyzed. The resulting spectrum (shown as the bottom trace) shows clear steps when the applied RF field (5 MHz) matched the transition frequency of one of the 3 isotopes contained in GaAs.

**5.8 Quadrupole interactions in semiconductors**

The combination of optical pumping with optical detection results in very high sensitivity, which allows one to record spectra with excellent signal to noise even from nanometer-sized structures such as quantum wells. At the same time, the optical excitation and detection allows one to distinguish the signal from these small structures from the much larger signals that the bulk





**Ideal GaAs crystal**
cubic symmetry
resonance lines are degenerate

**Strain, fields**
breaking of symmetry
non degenerate resonance lines

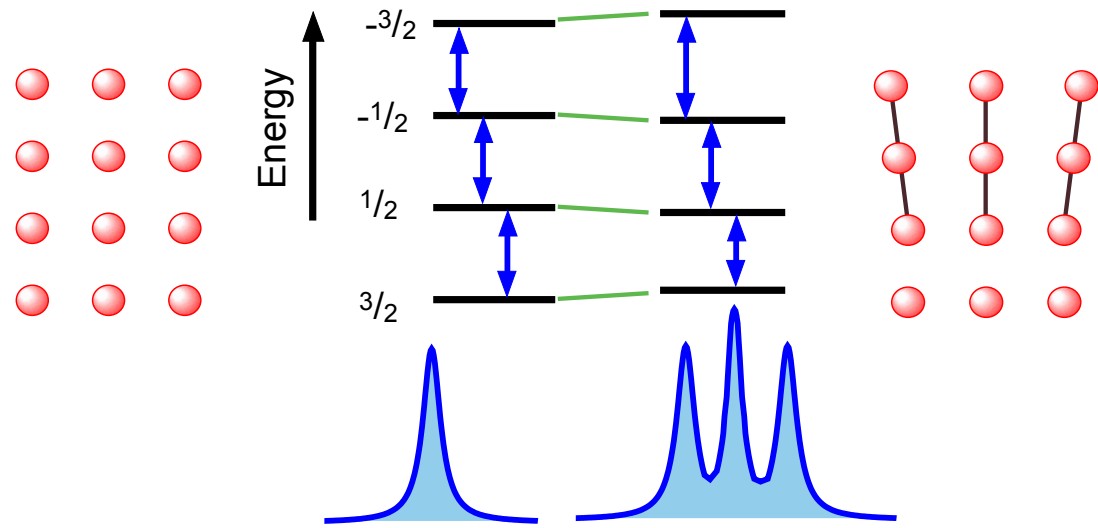

**Figure 27.** Spectra of $I = 3/2$ spins in ideal and distorted crystals.

material generates. One interesting application is to study the distortions that are generated at small scales by nanostructures like quantum wells or quantum dots, but also on larger scales by the effect of electric fields or mechanical strain.

Figure 27 shows the basic principle for the example of a nuclear spins $I = 3/2$, which corresponds to all 3 isotopes of Ga and As contained in GaAs. In an ideal GaAs crystal, the cubic site symmetry at the Ga and As sites assures that there is no electric field gradient (EFG) and the energy differences between the 4 spin states are identical, as they are split only by the Zeeman interaction. However, strain or electric fields, either external ones or fields generated by space charges in the material, can break the symmetry. In those cases, the EFG becomes nonzero and the quadrupole interaction lifts the degeneracy of the magnetic dipole transitions, as shown in the right-hand part of figure 27.

Figure 28 shows an example of an NMR signal that was obtained by scanning the RF frequency while the sample was subject to a constant magnetic field of 0.86 T. The clearly distinguishable steps in the change of the optical signal represent the three allowed transitions in a spin-3/2 system, as shown schematically in the right-hand part. The two satellite transitions are broader than the central transition, which is not affected by first-order quadrupole interaction.

Since all three isotopes of GaAs have spin $I = 3/2$, they all show the same type of quadrupole splitting, although to different degrees. Figure 29 compares the quadrupole splitting of all three isotopes, measured at the same location of the sample. [75]As shows the largest splitting, while the splitting of the two Ga isotopes is about a factor of 5 smaller. In parallel to the larger





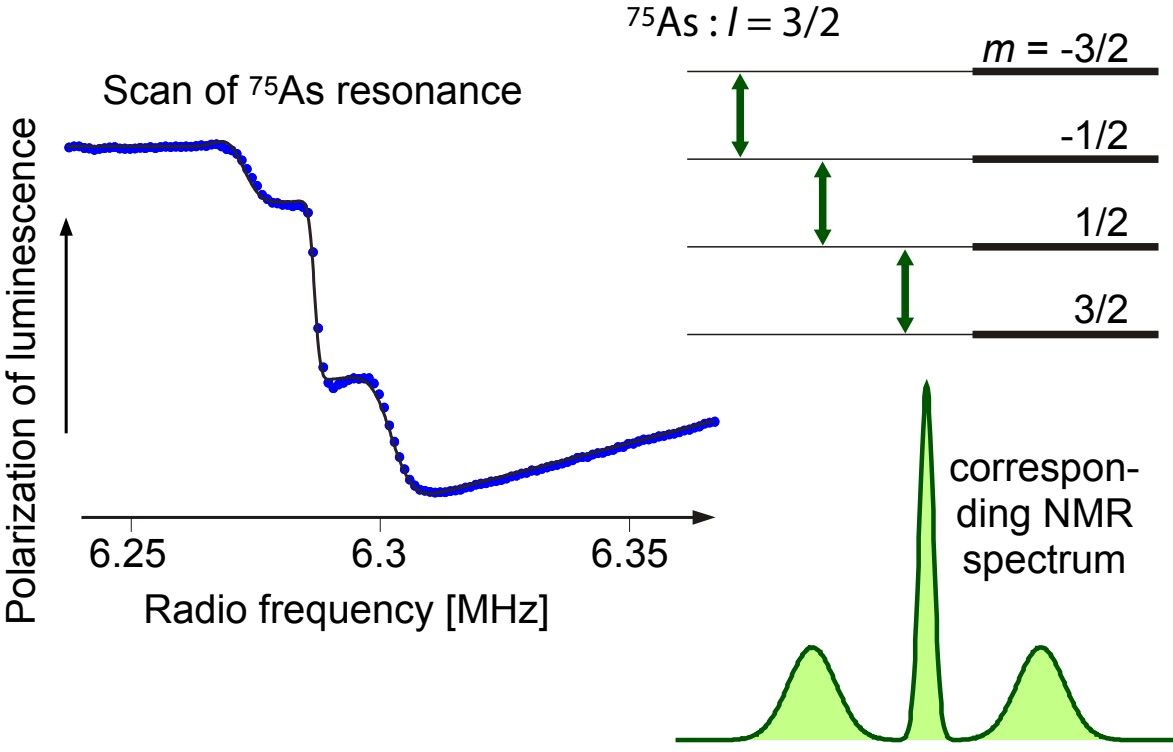

**Figure 28.** CW NMR spectrum measured by scanning the RF frequency in a constant field of 0.86 T.

quadrupole interaction, the As spectrum shows stronger broadening of the satellite lines, in agreement with expectations for inhomogeneous strain fields.

The high sensitivity of optical techniques as well as the possibility of using optical excitation to select specific parts of a sample make such experiments highly useful for studying so-called quantum-confined structures. Figure 30 shows the measured

quadrupole splittings in a sample containing 5 quantum wells with different thickness. The variation of the quadrupole splitting with depth can be explained (Eickhoff et al., 2003) through the charge distribution in a crystal when two different materials are in contact with each other. This is known as the Schottky-effect. Similar experiments have also been performed on different materials like Si, Ge (Glaser et al., 1990) and InSb (Hofmann et al., 1993a, b).

### 5.9  All-optical excitation and detection of NMR

Most experiments discussed here rely on classical radio-frequency fields to drive the nuclear spin transitions. However, it is also possible to perform experiments purely optically, with no RF irradiation but relying instead on the driving force generated by the optically excited charge carriers (Eickhoff et al., 2002). While this interaction is generally quite weak, it can be resonantly enhanced by modulating the amplitude or the polarisation of the laser field with a frequency near a transition frequency. Both types of modulation act on the nuclear spin via the hyperfine interaction, but the amplitude modulation additionally




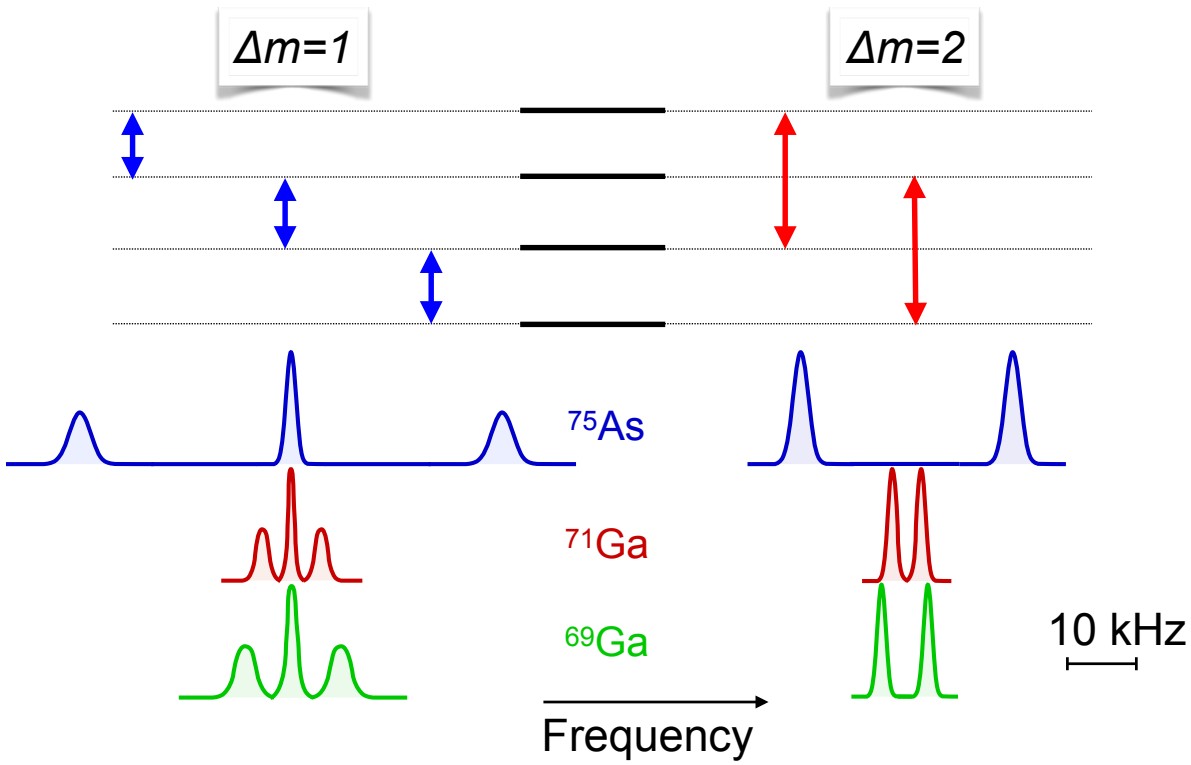

**Figure 29.** Quadrupole-split spectra of all 3 isotopes in the same EFG.

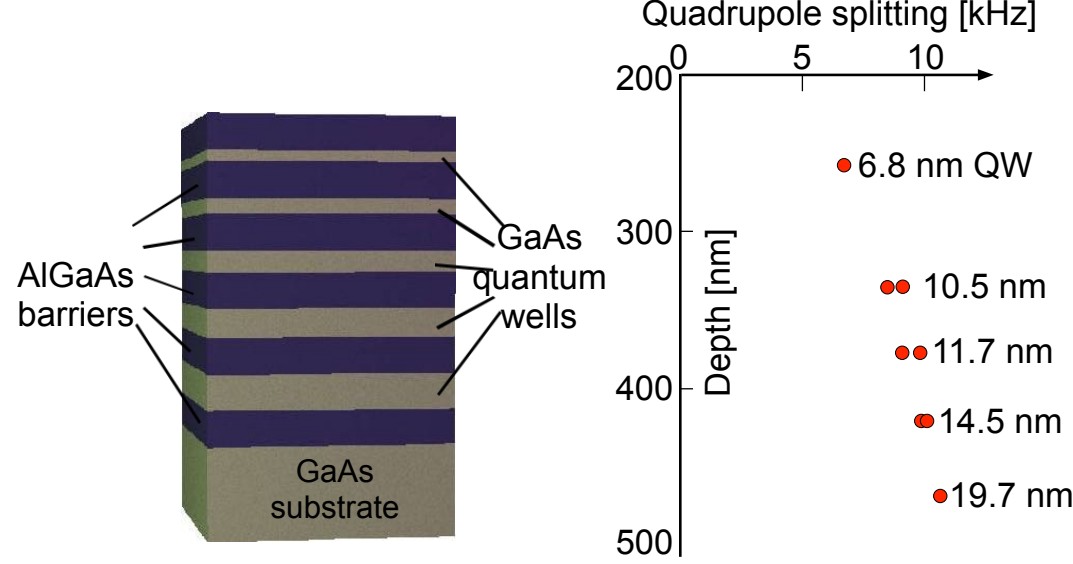

**Figure 30.** Variation of the quadrupole splitting with the distance from the sample surface.



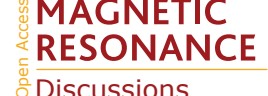

**Figure 31.** Spectra of single- and double-quantum transitions in GaAs, measured by modulated optical excitation. In the left-hand column, the polarisation of the laser beam was modulated, for the right-hand side, the amplitude. It therefore excites directly the $\Delta m = \pm 2$ transitions.





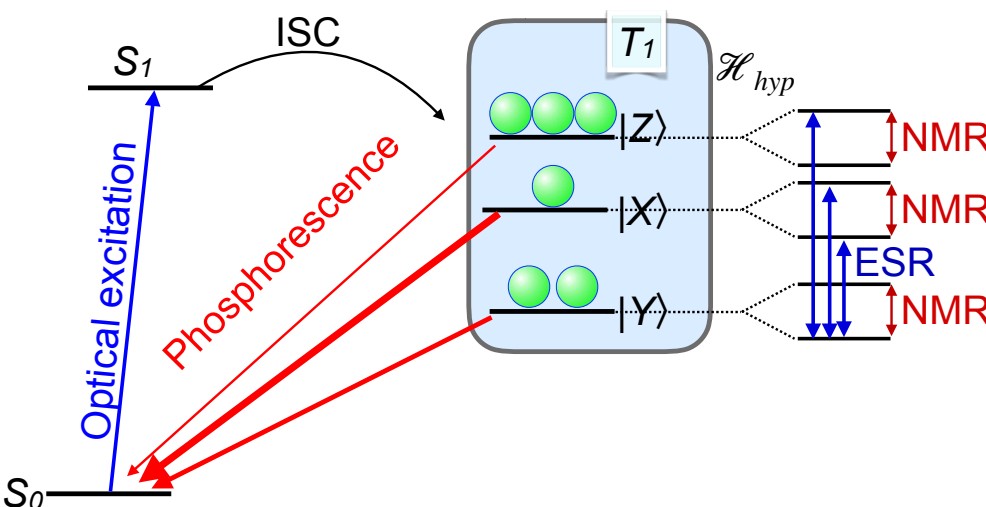

**Figure 32.** Optical excitation and detection of ESR and NMR transitions in a molecule with a triplet state.

modulates the charge distribution and therefore the electric field gradient at the nuclear spins. This can be used, e.g., to directly drive double quantum transitions, as shown in fig. 31. In this case, the modulation frequency has to satisfy the condition $\omega_{mod} = 2\omega_L$, where $\omega_L$ is the Larmor frequency of the targeted spin. This is fundamentally different from the case of 'normal' double quantum excitation with radio-frequency fields, where the condition is $\omega_{rf} = \omega_L$. The experimental data (dots), which correspond to the polarisation of the PL, are compared to a simulation (curve), which was calculated for the NMR spectrum

shown in the inserts of fig. 31.

## 6   Molecular solids

Conventional magnetic resonance has been applied most successfully to molecular systems. These systems can also be investigated with optical methods. Aromatic organic molecules are particularly suitable because their chromophores can support strong optical transitions.

Experiments with these systems typically use optical excitation from the ground state to an excited singlet state with a pulsed UV laser, as shown in figure 32. From the excited singlet state, intersystem crossing (ISC) can populate a nearby triplet state. The ISC populates the different levels of the triplet manifold unequally. In addition, the states of the triplet manifold have in general different lifetimes. As a result, the three different sublevels can have very different populations, as shown schematically in figure 32. The polarization and intensity of the phosphorescence originating from these states depends on the population of

the individual states and allows an indirect measurement of the population differences.

By applying microwave fields to the system, it is possible to induce transitions between the different triplet states and thereby change the rate of emitted photons. Accordingly, the transitions can be observed in the photoluminescence. If an appropriate





filtering procedure is implemented, most of the collected photons originate from molecules with a high spin polarisation, This makes it possible to detect magnetic resonance of individual molecules, such as pentacene in p-terphenyl hosts (Köhler et al., 1993; Wrachtrup et al., 1993). Initial experiments detected transitions between electron spin states, but soon afterward, also nuclear spin transitions in single molecules could be measured (Wrachtrup et al., 1997; Wrachtrup and Finkler, 2016).

As shown in the right-hand part of figure 32, the nuclear spin leads to an additional splitting. The non-thermal populations of the triplet states also leads to nuclear spin polarisation, which can survive the return to the ground state. The transfer of polarisation from the electron to the nuclei occurs spontaneously through the hyperfine interaction but can also be induced by microwave irradiation, as in DNP. Adding a radio-frequency field and scanning it over the relevant frequency range then results in optically detected ENDOR or ODENDOR (Crookham et al., 1992; Koschnick et al., 1996; Glaser et al., 1998; Wrachtrup et al., 1997). Alternatively, the enhanced nuclear spin polarisation can be used by conventional NMR of the ground state.

## 7 Conclusion and Outlook

Conventional magnetic resonance uses static and alternating magnetic fields for the study of ensembles of electronic and nuclear spins. This review covers an extension where optical fields are used as an additional tool. These optical fields are usually derived from a laser or they represent luminescence emitted by the sample. The main motivation for such experiments is, in most cases, the increased information content of such double resonance experiments, or the increase in sensitivity, which allows sometimes experiments with single spins. In other cases, the correlation between the optical and magnetic resonance degrees of freedom allow one to focus on specific parts of a sample or identify units that can not be uniquely identified from either modality alone.

While most of the basic principles required for this work have been known for a long time, the actual implementations only became possible through progress in different fields. Major examples are the introduction of tuneable and narrowband laser sources, modulators for light and sensitive detectors, which are now limited only by quantum mechanics. Applying these new technologies to physical systems was often pioneered by people working in fields like optics (classical or quantum-), molecular or solid state physics. Conversely, the results of such studies often provide highly valuable information for those fields. Excellent examples for this type of benefits can be found in the fields of semiconductor- and surface physics. Currently, the field is still evolving rapidly and it appears highly probably that new types of applications will be developed for the foreseeable future.

## 8 Acknowledgment

This review draws mostly from work in which I was involved during the past 30 years, together with many colleagues and friends in Dortmund and around the world. While many of them deserve to be mentioned here, I would like to minimise the list by mentioning only two people: Sophia Hayes who participated in some of that work and convinced me to accept the invitation, and Geoffrey Bodenhausen, who suggested that I write this review and helped shape it, e.g. by reading an



early version and providing extremely helpful feedback. Financial support for this work came many funding sources, but
mostly from the DFG, the SNF and the EU through various granting schemes. Specifically, for the publication of this article,
we acknowledge financial support by Deutsche Forschungsgemeinschaft and TU Dortmund University within the funding
programme Open Access Publishing.




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
