# Peer review of "Optical Detection of Magnetic Resonance"

_Magnetic Resonance, 2020_

## Referee Comment (RC1) · Anonymous Referee #1 · 10 May 2020

The manuscript is a comprehensive discussion on the topic of optical detection of magnetic resonance, a field that has picked up pace in the last decade. The article covers the progress in the field since its inception in 1970's. It would provide a very good starting point for anyone trying to get introduced to this emerging field. The overview of physics as well as techniques that are involved in the optical detection have been presented.

The author makes the connection between atomic systems that are usually probed optically and nuclear systems probed in rf/mw early in the article and takes a unified approach to explaining the electronic and nuclear magnetic resonance. The article starts with basic background material, like selection rules for allowed transitions between spin states and explains the ways of using light for creating and detecting spin polarisation in systems. Section 2 is dedicated to explaining the the physical processes involved in the creation and observation of the polarisation phenomenologically, very

basic mathematics is all that is used to elucidate the mechanisms. A consolidated view of how most of systems can be selectively pumped to a internal spin state using angular momentum conservation rules thus creating hyperpolarisation is shown.

The different systems in which magnetic resonance can be studied optically are addressed in detail in the rest of the paper. Section 3 concentrates on atomic and molecular systems, typically vapours, where both electronic as well as nuclear spins can be observed. Section 4 deals with solid state systems ranging from rare earth ions and transition metals to charge vacancy centers. Studies on semiconductor materials is detailed in section 5, where more physical processes specific to band gap structures are also introduced. This section gives details of specific systems and problems in semiconductor physics where optical detection of magnetic resonance can complement established methods of characterisation of material properties. Molecular solids are also briefly talked about. All the sections are concise, giving references to important break thorough and milestones achieved, thus making the compilation complete in this manuscript.

The article presents a intelligible and thorough summary of emerging and established techniques in optical magnetic resonance detection. There are a couple of points I would like to make that could perhaps help improve the readability of the discussion. First is regarding the section on coherent Raman scattering (section 2.4.3). The three level 'Lambda' system is perfect example for this, and it would add to the discussion to point out that since this is a coherent process and not spontaneous, the linewidths can be much narrower overcoming spontaneous relaxation rates. Second comment is to do with the the topic of Hanle effect in section 5.2. While here the example if that of a semiconductor system, the effect itself has been studied extensively in gaseous systems as well. The explanations themselves are general and can be used independent of the system, it would be worth while to point out the vapour magnetometers based on Hanle effect where one of the earliest to be commercialised and in fact the effect has been applied in stellar spectroscopy to elucidate astrophysical magnetic fields.

Perhaps this topic could be part of section 2?

---

## Referee Comment (RC2) · Kazuyuki Takeda (Referee) · 13 May 2020

This is a very instructive review manuscript, which would help the interested readers (I believe there would be many in the magnetic resonance community) to take the bird's-eye view over the topics in magnetic resonance where the optical fields plays important roles both in polarizing spin systems and in offering superior detection sensitivity. I have just a few comments below, and I believe the manuscript, upon minor revision, would be suitable for publication in MR.

- Regarding spin selective intersystem crossing (ISC) that polarizes the triplet state of pentacene, the rather new references are cited in lines 162-163 (Kothe 2010, Iinuma, 2000). However, the first sign of high electron polarization in pentacene was suggested (A.J. van Strien, J. Schmidt, Chem. Phys. Lett. 70 (1980) 513-517) and precise determination of polarization by transient ESR was reported (D.J. Sloop, H. Yu, T. Lin,

S.I. Weissman, J. Chem. Phys. 75 (1981) 3746-3757) much earlier. It would be nice to cite these papers as well.

- Fig.20 nicely demonstrates the asymmetric spectrum reflecting the population bias when the spin system is polarized and excited by a small-tip-angle pulse. Here, the spin temperature was estimated to be 1.2 mK, but would the author be able to discuss the population distribution over the four levels in this experiment? At least the static field and the carrier frequency should be informed, so that the readers can have a feeling of how high the polarization was in this example.

minor points:

- line 20: "population ratio" (close to zero, I think) should be read as "polarization" (close to unity).

- line 129: nthe -> the

―――――――――――――――――――――

---

## Author Comment (AC1) · 29 May 2020

I would like to thank both reviewers for their very positive and useful comments! All their suggestions are well justified and I will be happy to implement this. I am confident that this will further improve the paper.

————————————————————

---

## Author Response (AR1)

```
 1  Anonymous Referee #1
 2
 3  First is regarding the section on coherent Raman scattering (section
 …  2.4.3). The three level 'Lambda' system is perfect example for this, and
 …  it would add to the discussion to point out that since this is a coherent
 …  process and not spontaneous, the linewidths can be much narrower
 …  overcoming spontaneous relaxation rates.
 4
 5     **  Good point! ii will be happy to add this.
 6
 7  Second comment is to do with the the topic of Hanle effect in section
 …  5.2. While here the example if that of a semiconductor system, the effect
 …  itself has been studied extensively in gaseous systems as well. The
 …  explanations themselves are general and can be used independent of the
 …  system, it would be worth while to point out the vapour magnetometers
 …  based on Hanle effect where one of the earliest to be commercialised and
 …  in fact the effect has been applied in stellar spectroscopy to elucidate
 …  astrophysical magnetic fields. Perhaps this topic could be part of
 …  section 2?
 8
 9     **  Good idea! I would suggest to move it to section 2, between the
 …  current sections 2.4.2 (spontaneous emission) and 2.4.3 (coherent Raman
 …  scattering)
10
11  Kazuyuki Takeda (Referee)
12  - Regarding spin selective intersystem crossing (ISC) that polarizes the
 …  triplet state of pentacene, the rather new references are cited in lines
 …  162-163 (Kothe 2010, Iinuma, 2000). However, the first sign of high
 …  electron polarization in pentacene was suggested (A.J. van Strien, J.
 …  Schmidt, Chem. Phys. Lett. 70 (1980) 513-517) and precise determination
 …  of polarization by transient ESR was reported (D.J. Sloop, H. Yu, T. Lin,
 …  S.I. Weissman, J. Chem. Phys. 75 (1981) 3746-3757) much earlier. It would
 …  be nice to cite these papers as well.
13
14     **  Thanks for this suggestion - I will be happy to add them.
15
16  - Fig.20 nicely demonstrates the asymmetric spectrum reflecting the
 …  population bias when the spin system is polarized and excited by a
 …  small-tip-angle pulse. Here, the spin temperature was estimated to be 1.2
 …  mK, but would the author be able to discuss the population distribution
 …  over the four levels in this experiment? At least the static field and
 …  the carrier frequency should be informed, so that the readers can have a
 …  feeling of how high the polarization was in this example.
17
18     **  Thanks for this suggestion - I will be happy to add this.
19
20  minor points:
21  - line 20: "population ratio" (close to zero, I think) should be read as
```

```
21… "polarization" (close to unity).
22  - line 129: nthe -> the
23
24      **  Thanks for pointing these out - I will certainly change this.
25
26
```

[revised manuscript text omitted]